# Main-chain engineering of polymer photocatalysts with hydrophilic non-conjugated segments for visible-light-driven hydrogen evolution

Chih-Li Chang [1,2], Wei-Cheng Lin[1], Li-Yu Ting[1], Chin-Hsuan Shih[3], Shih-Yuan Chen [4], Tse-Fu Huang[1], Hiroyuki Tateno [4], Jayachandran Jayakumar[1], Wen-Yang Jao[1], Chen-Wei Tai[1], Che-Yi Chu[5], Chin-Wen Chen [6], Chi-Hua Yu [3], Yu-Jung Lu [2], Chi-Chang Hu [1], Ahmed M. Elewa [1], Takehisa Mochizuki[4] & Ho-Hsiu Chou [1] ✉

Photocatalytic water splitting is attracting considerable interest because it enables the conversion of solar energy into hydrogen for use as a zero-emission fuel or chemical feedstock. Herein, we present a universal approach for inserting hydrophilic non-conjugated segments into the main-chain of conjugated polymers to produce a series of discontinuously conjugated polymer photocatalysts. Water can effectively be brought into the interior through these hydrophilic non-conjugated segments, resulting in effective water/polymer interfaces inside the bulk discontinuously conjugated polymers in both thin-film and solution. Discontinuously conjugated polymer with 10 mol% hexaethylene glycol-based hydrophilic segments achieves an apparent quantum yield of 17.82% under 460 nm monochromatic light irradiation in solution and a hydrogen evolution rate of 16.8 mmol m$^{-2}$ h$^{-1}$ in thin-film. Molecular dynamics simulations show a trend similar to that in experiments, corroborating that main-chain engineering increases the possibility of a water/polymer interaction. By introducing non-conjugated hydrophilic segments, the effective conjugation length is not altered, allowing discontinuously conjugated polymers to remain efficient photocatalysis.

Inspired by natural photosynthesis, researchers are developing photocatalysts to efficiently convert abundant solar energy into low-carbon chemicals and fuels. Hydrogen is an attractive primary energy source for a carbon-neutral society because it produces only water and heat during combustion[1–3]. Therefore, photocatalytic water splitting to produce zero-emission hydrogen has attracted considerable attention in the effort to address global challenges related to energy crises and environmental pollution[4–6]. However, the development of efficient and stable photocatalysts that can suppress electron-hole recombination and facilitate electron transfer in the photocatalytic reaction remains a challenge because it requires well-managed light harvesting and effective engineering of the energy levels, photocatalytic interface, and reaction mechanism. Thus, it is imperative to develop a strategy for the systematic study of photocatalysts with precise control of their molecular structure.

In recent decades, most studies have focused on the fabrication of inorganic photocatalysts, particularly metal-based semiconductors, which have low photocatalytic activity in the visible region, require

**Fig. 1 | Schematic illustration of design strategy and polymer structures.**
**a** Schematic illustration of the polymer photocatalysts containing hydrophilic non-conjugated segments. **b** The molecular structures of hydrophilic segments and the corresponding DCPs.

harsh fabrication conditions, and are difficult to optimize[7]. In 1985, Yanagida et al. first reported poly(p-phenylene) as an organic photocatalyst for hydrogen evolution during water splitting[8]. Since then, conjugated polymers (CPs) have attracted considerable attention[9–15]. Nevertheless, the main limitation of CPs with a hydrophobic skeleton is their nonhomogeneous dispersion under the conditions used for water-based reactions, where the use of an organic co-solvent is necessary. In addition, the severe aggregation of polymer chains and the poor water/polymer interfacial behavior limit electron transfer through the interface, and thus suppress hydrogen production via a two-electron process. Therefore, organic polymer photocatalysts with suitable hydrophilicity are potential candidates for water splitting with less/no organic co-solvent. Recently, many groups and us demonstrated that the hydrogen evolution rate (HER) of hydrophobic CP photocatalysts could be significantly increased under water-based photocatalytic conditions when covered with amphipathic small molecules or polymer surfactants[16–22].

In another strategy, hydrophilic functional groups (e.g., ethylene glycol, carboxylic acid, amino and ionic electrolyte groups) are incorporated on the side-chain of hydrophobic CPs to intrinsically improve the hydrophilicity of the CPs[23–26]. Although all previous studies introduced such hydrophilic groups on the side-chains of CPs, the backbones of side-chain-engineered CPs are hydrophobic, similar to those of conventional CPs. These previous studies considered that a long conjugation length was necessary for polymer photocatalysts to achieve better charge transport; however, the ideal effective conjugation length for photocatalytic hydrogen evolution is still unclear. The interior of the bulk side-chain-engineered CPs was still affected by the same issue as conventional CPs, resulting in intense electron-hole recombination and a poor water/polymer interface. Moreover, side-chain engineering is restricted to specific structures and results in poor applicability. Herein, main-chain-engineered DCPs were achieved through a delicate balance of hydrophilic non-conjugated segments and hydrophobic conjugated segments, resulting in an enhanced water/polymer interface to facilitate photocatalytic hydrogen evolution (Fig. 1a). Importantly, the types and contents of hydrophilic non-conjugated segments can be easily tuned and polymerized with various conjugated segments, which shows higher possibility and flexibility in our main-chain-engineered strategy.

## Results
### Polymer synthesis and characterization
To demonstrate our design strategy, poly[(9,9-dioctyl-9H-fluorenyl-2,7-diyl)-co-(5-phenylbenzo[b]phosphindole-5-oxide-2,7-diyl)]

(PFBPO) with a hydrophobic backbone was chosen as the reference CP for comparison with the DCPs because of its relatively high HER in visible-light-driven photocatalytic reactions[27]. We designed ethylene glycol (EG)- and ethylene diamine (EA)-based hydrophilic non-conjugated segments and incorporated them into the backbone of PFBPO. We synthesized EG- and EA-modified DCPs via Pd-catalyzed Suzuki−Miyaura coupling polymerization of fluorene-boronic ester and 3,7-dibromo-5-phenylbenzo[b]phosphindole-5-oxide with either EG-based (EG-Br, TEG-Br, and HEG-Br) or EA-based (EA-Br) segments. The hydrophilic non-conjugated segments were covalently bonded to the backbone of PFBPO at various ratios (5, 10, and 20 mol%), resulting in seven DCP photocatalysts, denoted P-EG-5, P-TEG-5, P-HEG-5, P-HEG-10, P-HEG-20, P-EA-5, and P-EA-10 (Fig. 1b and Supplementary Fig. 1). First, the monomers were synthesized and their chemical structures were identified by NMR spectroscopy and mass spectrometry. The monomers were then polymerized into the corresponding EG- and EA-modified DCPs, the chemical structures of which were investigated in detail using NMR and FT-IR spectroscopy. The $^1$H NMR spectra showed characteristic signals of the EG- and EA-based segments at 3.5–4.5 ppm, indicating that hydrophilic non-conjugated groups were present in the polymer backbones. The intensities of the characteristic signals of HEG-Br and EA-Br in the DCP backbones clearly increased with increasing molar fractions of the HEG-Br and EA-Br monomers during polymerization (Supplementary Fig. 2). Characteristic NMR and FT-IR signals of the phosphine oxide groups were observed at 33.80 ppm and 1140–1210 cm$^{-1}$ (Supplementary Fig. 3), respectively. The molecular weight and polydispersity index of the DCPs were determined using gel permeation chromatography (Supplementary Table 1). The results of thermogravimetric and X-ray diffraction analysis indicated that all prepared polymers had an amorphous framework that was highly stable in a nitrogen atmosphere ($T_d$ > 400 °C; see Supplementary Figs. 4 and 5).

### Optical and electrochemical properties
The optical properties, optical band gap ($E_g$), and band structure of the DCPs were studied by diffuse reflectance UV−vis spectroscopy (DRS), photoluminescence spectroscopy, and photoelectronic spectroscopy. The DRS of the DCPs was very similar to that of PFBPO (Fig. 2a). Their highest occupied molecular orbital (HOMO) levels ranged from −5.90 to −5.93 eV, as measured using a photoelectronic spectrometer (Supplementary Fig. 6). Furthermore, the oxidation potential of these polymers measured using cyclic voltammetry (CV) exhibited comparable results to photoelectronic spectroscopy (Supplementary

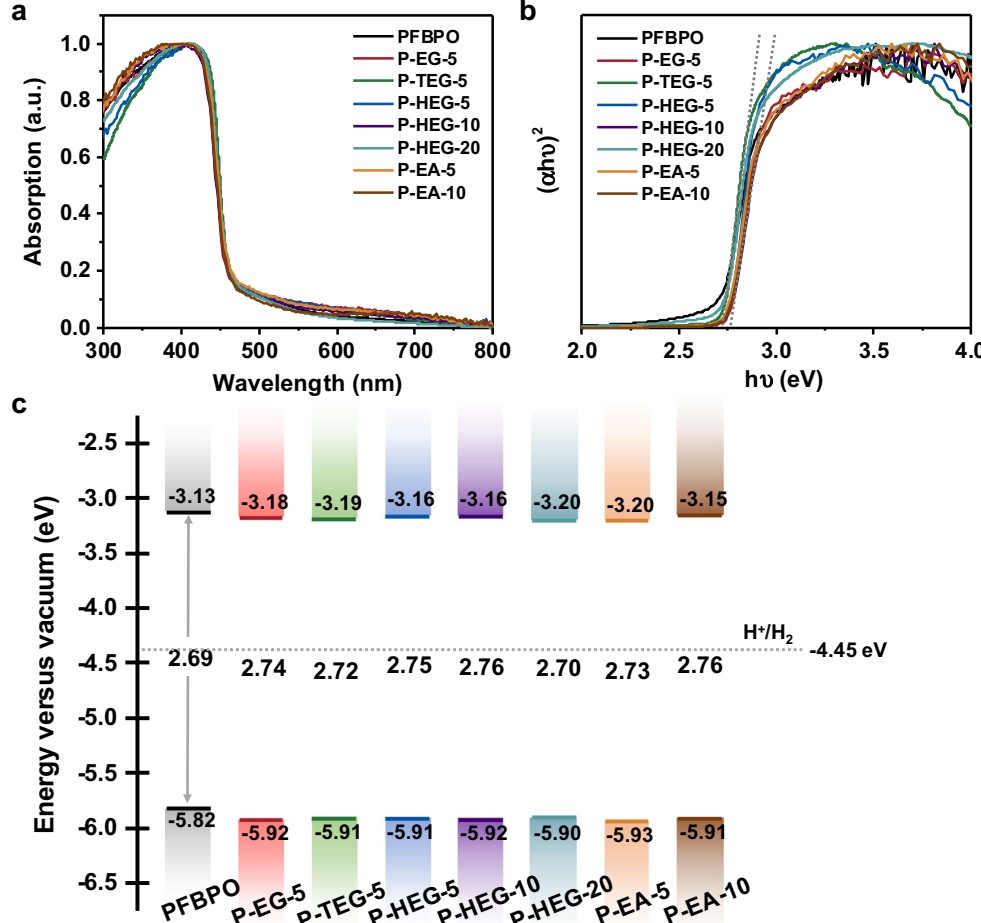

**Fig. 2 | Optical properties of polymers. a** Solid-state UV-vis diffuse reflectance spectra, **b** Tauc plots, and **c** energy level diagram of all polymers as measured by PESA in conjunction with Tauc plot. The dashed lines correspond to the proton reduction potential (H⁺/H₂). All energy levels and electrochemical potentials are expressed relative to vacuum (using −4.44 V versus vacuum as equivalent to 0 V versus SHE)[32]. Energy levels measured by cyclic voltammetry (Supplementary Fig. 7) in conjunction with Tauc plot indicate slightly different energy levels. However, in both cases the energy levels are suitable for proton reduction.

Fig. 7). The lowest unoccupied molecular orbital (LUMO) levels were calculated using $E_{LUMO} = E_{HOMO} + E_g$, where the $E_g$ values of the DCPs calculated from Tauc plots were close to 2.75 eV (Fig. 2b); their LUMO levels ranged from −3.15 to −3.20 eV, indicating that hydrogen evolution was thermodynamically favorable (Fig. 2c and Table 1). As shown in Fig. 3a, the emission maxima of all polymers fall at almost the same wavelength. Furthermore, all polymers were effectively quenched by the addition of Pt co-catalyst (Supplementary Fig. 8). The photoluminescence quenching degree of DCPs was higher than that of PFBPO, indicating improved charge transfer from DCPs to the Pt co-catalyst. Time-resolved photoluminescence spectroscopy including instrument response function curves (Fig. 3b and Supplementary Fig. 9) shows that there is no significant difference in exciton lifetimes between all polymers, with their excited-state lifetimes ranging from 0.85 to 1.07 ns. To sum up, the HOMO/LUMO levels, optical $E_g$, photoluminescence spectra, and exciton lifetime of all polymers showed no obvious difference, indicating that hydrophilic non-conjugated segments could be molecularly engineered into the main-chain of the target CPs using the present approach without significantly affecting the physicochemical and optical properties.

Electron paramagnetic resonance (EPR), electrochemical impedance spectroscopy (EIS), and transient photocurrent measurements were used to characterize the charge generation, transport, and transfer upon irradiation of the DCPs. In the EPR spectra (Fig. 3c and Supplementary Fig. 10), a sharp signal was observed at 3500–3520G

and its intensity increased with additional light irradiation due to more radical generation via additional photoexcitation. The irradiation-induced signal enhancement for the DCPs was greater than that for PFBPO. Interestingly, after P-HEG-10 was stored in the dark for 3 days, although the EPR signal could still be observed, the intensity of the EPR signal was significantly decreased. Therefore, polymers already showing EPR signals without additional light irradiation can be attributed to their responded to light when stored under ambient condition. In addition, we further measured the EPR of the monomers (Supplementary Fig. 11), and found that the EPR signal of the polymer was mainly contributed by the BPO segment instead of the fluorene and HEG segments. The EIS data recorded in the dark showed a semi-circular curve (Fig. 3d and Supplementary Fig. 12), where the charge transfer resistance values of DCPs is lower than that of PFBPO due to the incorporation of hydrophilic non-conjugated segments, suggesting that the DCPs have a more suitable interface for charge transfer in photocatalytic hydrogen production reaction. Transient photocurrent measurements used to investigate the photoresponse of all polymer. After fabricating the polymer film, we measured the thickness of polymer film using dual beam-focused ion beam (DB-FIB), and the thickness range was ~571–585 nm (Supplementary Fig. 13), demonstrating that all polymer films have comparable thickness to each other. As results, the DCPs showed that these materials responded quickly to light irradiation and their photocurrents were higher than that of PFBPO (Fig. 3e).

**Table 1 | Photophysical properties and HERs of various polymers**

| Polymer | $\lambda_{max,\ abs}$ (nm)[a] | Size (nm)[b] | HOMO (eV)[c] | LUMO (eV)[d] | $E_g$ (eV)[e] | Lifetime (ns)[f] | HER 780 > λ > 380 nm ($\mu$mol h$^{-1}$)[g] | HER 780 > λ > 380 nm (mmol h$^{-1}$ g$^{-1}$)[g] |
|---|---|---|---|---|---|---|---|---|
| PFBPO | 410 | 471 | −5.82 | −3.13 | 2.69 | 0.85 ± 0.011 | 23.0 ± 1.85 | 4.60 ± 0.37 |
| P-EG-5 | 410 | 654 | −5.92 | −3.18 | 2.74 | 0.87 ± 0.011 | 23.9 ± 2.60 | 4.78 ± 0.52 |
| P-TEG-5 | 408 | 674 | −5.91 | −3.19 | 2.72 | 0.90 ± 0.012 | 25.8 ± 1.35 | 5.16 ± 0.27 |
| P-HEG-5 | 414 | 686 | −5.91 | −3.16 | 2.75 | 0.95 ± 0.012 | 26.9 ± 1.55 | 5.38 ± 0.31 |
| P-HEG-10 | 402 | 1129 | −5.92 | −3.16 | 2.76 | 1.02 ± 0.017 | 34.7 ± 3.45 | 6.94 ± 0.69 |
| P-HEG-20 | 390 | 1576 | −5.90 | −3.20 | 2.70 | 1.07 ± 0.011 | 18.7 ± 1.80 | 3.74 ± 0.36 |
| P-EA-5 | 402 | 957 | −5.93 | −3.20 | 2.73 | 0.93 ± 0.012 | 30.3 ± 3.05 | 6.06 ± 0.61 |
| P-EA-10 | 396 | 1434 | −5.91 | −3.15 | 2.76 | 0.98 ± 0.011 | 19.4 ± 1.95 | 3.88 ± 0.39 |

[a]Determined by solid-state UV-vis diffuse reflectance spectroscopy.
[b]Determined by DLS in a solution mixture consisting of equal volumes of $H_2O$, MeOH, and TEA.
[c]Determined by photoelectron spectroscopy.
[d]Determined by means of the equation $E_{LUMO} = E_{HOMO} + E_g$.
[e]Determined from Tauc plots.
[f]Determined by time-resolved fluorescence spectroscopy in a solution mixture consisting of equal volumes of $H_2O$, MeOH, and TEA.
[g]Photocatalytic conditions: 5 mg of polymers dissolved in a 10 mL solution mixture consisting of equal volumes of $H_2O$, MeOH, and TEA and irradiated by a simulated solar light source (300 W Xe lamp, AM 1.5, 1000 W m$^{-2}$, 780 > λ > 380 nm).

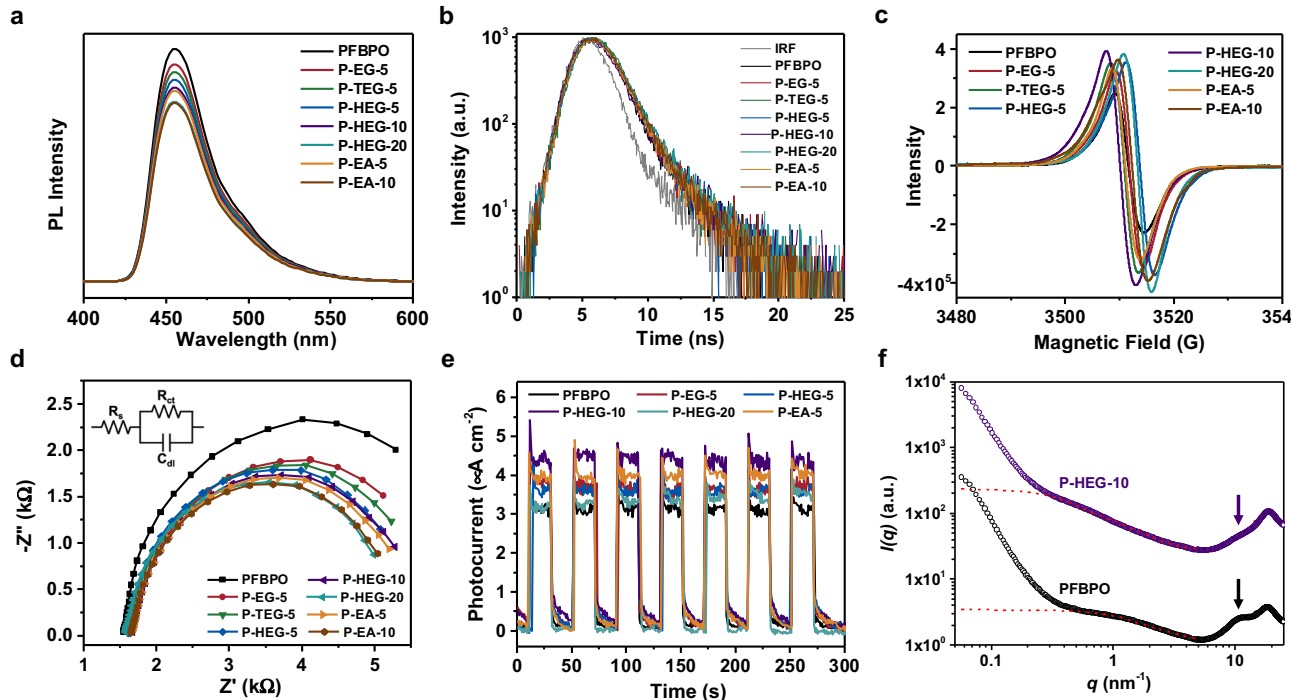

**Fig. 3 | Optical and electrochemical properties of polymers.**
**a** Photoluminescence spectra were obtained in suspensions (5 mg photocatalyst in 10 mL solution mixture consisting of equal volumes of $H_2O$, MeOH, and TEA). **b** Time-resolved photoluminescence spectra were obtained in suspensions (5 mg photocatalyst in 10 mL solution mixture consisting of equal volumes of $H_2O$, MeOH, and TEA). Samples were excited with a 405 nm laser and emission was measured at 455 nm. **c** EPR spectra were collected in the solid state under additional light illumination. **d** Electrochemical impedance spectra of polymers were carried out in dark with an AC potential frequency ranging from 0.1 Hz to 100 kHz. In the equivalent circuit, $R_s$ represents the circuit series-resistance, $R_{ct}$ is the charge transfer resistance across the interface, and $C_{dl}$ is the capacitance phase element of the semiconductor-electrolyte interface. **e** Photocurrent were generated upon light on-off switching. **f** Wide-angle X-ray scattering profiles of PFBPO and P-HEG-10 were measured in a mixture solution consisting of equal volumes of $H_2O$, MeOH, and TEA.

## Morphology and hydrophilicity

Scanning electron microscopy (SEM; Supplementary Figs. 14–21) images showed that the solid DCPs consisted of small particles with similar morphologies and microscale sizes, independent of the hydrophilic non-conjugated segments. The water contact angles of the EG-Br, TEG-Br, HEG-Br, and EA-Br monomers were 95.7°, 82.1°, 47.4°, and 76.7°, respectively (Supplementary Fig. 22), where HEG-Br had the highest hydrophilicity (lowest contact angle). Interestingly, we measured the contact angles of water drop on all of the DCPs films

promptly, and observed hydrophobic behavior with the contact angles of ~109° (Supplementary Fig. 23), which were in contrast to most side-chain-engineered CPs that show hydrophilic behavior[19–22]. We considered that the high density packing of polymer chains in the films results in non-obvious differences in water contact angles between DCPs and PFBPO. To further understand the difference of the hydrophilicity between the DCPs and PFBPO, the particle size distributions of the polymers in a non-aqueous (toluene), aqueous (pure water), and mixture solution (33.3 vol.% $H_2O$/33.3 vol.% MeOH/33.3 vol.% TEA)

were determined by dynamic light scattering (DLS). As shown in Supplementary Fig. 24, all of the DCPs showed smaller hydrodynamic diameters than PFBPO in toluene, indicating that the DCPs have higher solubility in non-aqueous solution. We considered it is because the insertion of the non-conjugated hydrophilic segments to the polymer backbone will reduce the packing of the polymers. In addition, the polymers showed larger hydrodynamic diameters in pure water (Supplementary Fig. 25) than in the mixture solutions (Supplementary Fig. 26), because there is no organic solvent to help the dissolution of the polymers. Importantly, in both pure water and mixture solutions, all the DCPs showed larger hydrodynamic diameters than PFBPO due to swelling phenomena in aqueous solution[28,29].

To further investigate the morphology of the DCPs, small-angle X-ray scattering (SAXS) measurements were performed by dispersing the DCPs in a $H_2O$/MeOH/TEA solution with a volume ratio of 1:1:1 and comparing their architectures. The intermediate- and high-$q$ scattering characteristics ($q = 0.4-4$ nm$^{-1}$) shown in the SAXS patterns of P-HEG-5, P-HEG-10, and P-HEG-20 (Supplementary Fig. 27) were ascribed to a one-dimensional rod geometry with polydispersity in the rod size[30]. Fitting of the high-$q$ scattering profile indicated that the DCPs formed rod-like bundles in aqueous solution and that their mean radius and mean length ($0.6 \pm 0.0002$ nm & $2 \pm 0.004$ nm) were independent of the content of hydrophilic non-conjugated segments. However, the low-$q$ scattering profile ($q < 0.4$ nm$^{-1}$) of P-HEG-10 was fitted with the lowest power law exponent of $-2.0$, revealing that the rod-shaped bundles of P-HEG-10 can be further organized into an aggregated structure with a random distribution. The lower degree of aggregation implied a loose arrangement of the polymer bundles, resulting in a larger interfacial area. Interesting, the level of hydration/swelling in P-HEG-10 could be evidenced by an obvious increase in the mean size of the rod-shaped bundles as resolved by the wide-range scattering profile covering both the SAXS and WAXS regions with compared to that of PFBPO (Fig. 3f). It was clearly shown that the form factor ($q = 0.3-4$ nm$^{-1}$) contributed from the rod-like bundles of P-HEG-10 located at the $q$ range lower than that of PFBPO ($q = 0.4-4$ nm$^{-1}$), indicating that the average size of the bundles formed in P-HEG-10 was larger than PFBPO in response to the effective expansion of intra-chain distance via hydration/swelling mediated by the hydrophilic segments. Comparison between the fitted curves of rod form factor scatterings revealed that the geometric sizes of P-HEG-10 bundles ($0.65$ nm $\pm 0.0002$ in radius and $11.1$ nm $\pm 0.004$ in length) were indeed larger than that of PFBPO bundles ($0.46$ nm $\pm 0.0002$ in radius and $3.4$ nm $\pm 0.004$ in length). Particularly, the size expansion of P-HEG-10 bundles in the radial direction seems to be beneficial to increase the length of the bundles. It could be deduced that absorbed water molecules inside the bundles were able to act as structure-directing agent to induce more chain segments to be involved in packing along the axial direction of the bundles. This shall enhance the specific surface area from the aligned polymer bundles for higher photocatalytic activity relative to the entangled state. A further investigation into the smearing of the diffraction at $q = 10$ nm$^{-1}$ for P-HEG-10 suggested a perturbation in the ordering of local chain association, which could also be attributed to the hydration/swelling that may locally inhibit the chain packing at atomic length scale.

## Photocatalytic hydrogen evolution

The photocatalytic hydrogen evolution of the DCPs was measured at 25 °C under visible-light irradiation (AM 1.5, $\lambda = 380-780$ nm) and compared to that of PFBPO. Mixtures of 33.3 vol.% $H_2O$, 33.3 vol.% MeOH, and 33.3 vol.% TEA were used, where TEA acts as the sacrificial electron donor and MeOH is added to aid mixing of the TEA with water. Figure 4a shows that the HER values of the DCPs with 5–10 mol% hydrophilic non-conjugated segments were clearly higher than that of PFBPO. In particular, a stable HER of P-HEG-10 was observed with increasing reaction time, and an excellent HER of 34.7 µmol h$^{-1}$ was

obtained, which was ~50% higher than that of PFBPO (23.0 µmol h$^{-1}$). The hydrophilic non-conjugated segments facilitated the swelling of the polymer aggregates in the presence of water and thus enhanced the HER of the DCPs (Table 1). However, the HER of P-HEG-20 (with 80 mol% BPO active segments and 20 mol% HEG hydrophilic segments) was lower, with a value of 18.7 µmol h$^{-1}$. Therefore, the optimal content of hydrophilic HEG segments on the backbone of DCPs was ~10 mol%. Considering the SAXS results, the fact that P-HEG-10 exhibited the highest HER was closely related to its loose aggregates, which provide a high interfacial area for hydrogen evolution.

The P-EA-5 sample also showed a high HER of 30.3 µmol h$^{-1}$, which was higher than those of its counterparts with 5 mol% EG-based hydrophilic segments (e.g., P-EG-5, P-TEG-5, and P-HEG-5). The EA-based hydrophilic segments with N-containing functional groups experienced stronger hydrogen bonding with water than the EG-based hydrophilic segments with oxygen-containing functional groups. In addition, when 10 mol% EA was present, the corresponding P-EA-10 exhibited a relatively low HER of 19.4 µmol h$^{-1}$. In the EA-based DCPs, the optimal content of EA hydrophilic segments on the backbones of the DCPs was ~5 mol%. As seen from the DLS data, P-HEG-10 and P-EA-5 were significantly swollen with water in the $H_2O$/MeOH/TEA solution, which were beneficial for water-based photocatalytic hydrogen evolution, forming aggregates with a diameter of ~1000 nm. On the other hand, the HER of P-HEG-20 and P-EA-10 show a lower HER than the hydrophobic PFBPO. We considered that it is owing to the multiple scattering effect and lower light absorption of P-20HEG and P-10EA[17]. In order to address this issue, as shown in Fig. 4b, we increased the concentrations of photocatalytic solution of P-HEG-20 and then compared with that of PFBPO. Notably, the result showed that the HER enhancement of P-HEG-10 and P-HEG-20 both are more significant than that of PFBPO under the concentration of 10 mg/10 mL and 15 mg/10 mL solution. (Fig. 4b and Supplementary Table 1). This finding suggests that the lower HER enhancement of PFBPO under the higher concentration also demonstrated that the aggregation of hydrophobic PFBPO was increased when the concentration was increased, so more of the inner polymer chains of the PFBPO cannot interact effectively with water. Excitingly, even though P-HEG-20 had a lower HER in a 5 mg/10 mL solution, P-HEG-20 exhibited a higher HER than PFBPO in the higher solution concentration.

Likewise, a linear increase in the HER was observed with increasing quantity of photocatalytic solutions in the absence of a Pt co-catalyst (Fig. 4c and Supplementary Fig. 28), demonstrating that, at a fixed concentration, the total amount of hydrogen production could be increased by increasing the reactor size. Apparent quantum yield (AQY) values were obtained under standard photocatalytic conditions using a light source with a bandpass filter ($\lambda = 420$, 460, 500, 550, or 600 nm). Without a Pt co-catalyst, P-HEG-10 exhibited high AQYs of 18.19% and 17.82% at 420 and 460 nm, respectively (Fig. 4d and Supplementary Table 2). The trend in AQY values as a function of wavelength was very similar to that of the UV-vis absorption spectra of P-HEG-10, indicating that photocatalytic hydrogen evolution occurred via light harvesting. As shown in Supplementary Fig. 29, the long-term photocatalytic cycling hydrogen evolution evidenced that no any distinct roll-off in the photocatalytic activity for P-HEG-10 during the continuous photocatalytic reaction for 48 h (8 cycles), indicating P-HEG-10 has impressive photo-stability and durability. Taking advantage of the strong interaction between the hydrophilic non-conjugated segments and the Pt co-catalyst, which can enhance the charge separation and transfer at the interfacial area[26,31], the HER of P-HEG-10 with 5 wt.% Pt co-catalyst was increased to 54.1 µmol h$^{-1}$ (10.82 mmol h$^{-1}$ g$^{-1}$; Supplementary Fig. 30).

We further verified the performance of the DCPs for photocatalytic hydrogen evolution in the absence of MeOH (i.e., in an 80 vol.% $H_2O$/20 vol.% TEA solution, Fig. 4e). PFBPO exhibited a very low HER (6.79 µmol h$^{-1}$), while P-HEG-10 and P-EA-5 exhibited values of up

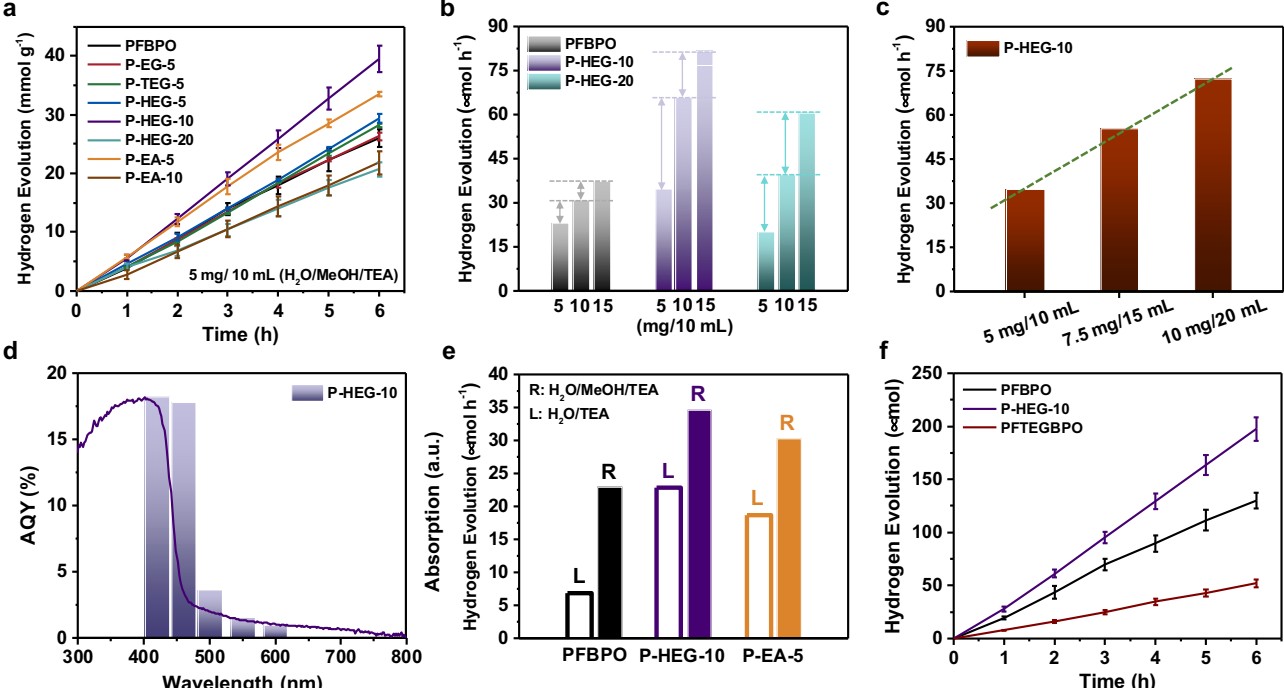

**Fig. 4 | Photocatalytic hydrogen evolution experiments in solution. a** Time-dependent HER of DCP and PFBPO photocatalysts under 380–780 nm irradiation (5 mg of polymer powder and 10 mL of a mixed solution consisting of 33.3 vol.% H₂O, 33.3 vol.% MeOH, and 33.3 vol.% TEA). **b** Concentration dependence of the HER over PFBPO, P-HEG-10, and P-HEG-20. **c** Linear relationship between the HER and the quantity of photocatalytic solution (10, 15, and 20 mL of solution with a constant P-HEG-10 concentration). **d** Correlation between the apparent quantum yield (AQY) and the UV-vis absorption spectra of P-HEG-10. **e** HER values in the absence of MeOH (i.e., in an 80 vol.% H₂O/20 vol.% TEA solution) over PFBPO, P-HEG-10, and P-EA-5 (L columns) compared to the HER values in presence of MeOH (i.e., in a 33.3 vol.% H₂O/33.3 vol.% MeOH/33.3 vol.% TEA solution) (R columns). **f** Comparison of the time-dependent HER over P-HEG-10, PFBPO, and PFTEGBPO using the same photocatalytic conditions as used in **a**.

to 22.8 µmol h⁻¹ and 18.6 µmol h⁻¹, respectively, in the H₂O/TEA solution, and these values were close to that of PFBPO in the presence of MeOH (23.0 µmol h⁻¹). Therefore, the introduction of 5–10 mol% HEG-Br or EA-Br hydrophilic segments on the main-chain of DCPs could replace the use of the typically used 33.3 vol.% MeOH. In addition, compared to side-chain engineering of CPs (containing 50 mol% hydrophilic segments), our approach required a smaller amount of hydrophilic segments (5–10 mol%) and could effectively increase the interaction between water and the inner active sites of the CPs to increase the HER. To further demonstrate the advantages and applicability of our design approach, we synthesized a side-chain-engineered CP (denoted as PFTEGBPO) to compare its photocatalytic HER with that of the P-HEG-10 DCPs (Fig. 4f). We considered it is because PFTEGBPO exhibits higher transmittance in the range of 380–780 nm, as shown in Supplementary Fig. 31, resulting in the inefficient light harvesting and poor HER performance. Furthermore, to demonstrate the universality of our approach, we extended the hydrophilic HEG-Br segment to polymerize with other published donor–acceptor CPs. PF8BT-HEG-10 and PCPDTBSO-HEG-10 samples were synthesized by the main-chain engineering of 10 mol% HEG-Br into the backbone of PF8BT and PCPDTBSO, both which exhibited enhanced HERs compared to that of PF8BT and PCPDTBSO, respectively (Supplementary Fig. 32). It can be seen that the absorption maxima of the polymers are red-shifted from 400 nm for P-HEG-10 to 450 nm for PF8BT-HEG-10 and then to 520 nm for PCPDTBSO-HEG-10, and the enhanced photocatalytic hydrogen evolution results are obtained in all three polymers. All these demonstrate that we can easily applied our method to the other polymer, which the absorption is located at visible-light region.

Furthermore, we coated the PFBPO and P-HEG-10 to obtain uniform films on silicon wafers by only single dropcast cycle (Fig. 5a, b). As shown in Fig. 5c and Supplementary Table 3, the P-HEG-10 showed an

excellent HER of 16.6 mmol m⁻² h⁻¹ in a film state. Interestingly, the HER of P-HEG-10 showed 2-times enhancement compared to that of PFBPO, which is higher than the enhancement in solution state (1.5-times). To further demonstrate the photocatalytic stability of the polymer thin-films, we directly soaked the PFBPO and P-HEG-10 thin-films in photocatalytic solution under light illustration for over 120 h. As shown in Supplementary Fig. 33, both the PFBPO and P-HEG-10 thin-films remain good quality even after 120 h, demonstrating the introduction of small fraction the hydrophilic segments into the polymer backbone did not obviously affect the photocatalytic stability of P-HEG-10 thin-film. We also measured the water contact angle of film photocatalysts over a period of time to determine the wetting effect of these films (Fig. 5d, e). In the case of the P-HEG-10 film with the initial contact angle of 107.9° and then decreased to 65.5° over the period of 20 min. While for the PFBPO film, the initial water contact angle of 112.0° decreased to 79.3° after 20 min. Since the polymer chains in the film state are molecularly packed at a high density, resulting in that the interior of the polymer film may not be able to efficiently react with water, this result demonstrated the advantage of our main-chain engineered DCP as a thin-film photocatalyst.

## Molecular dynamics simulation
Molecular dynamics simulation provides a microscale understanding of the interaction between water and DCP (Fig. 6a). Improvement in hydrogen evolution results from a higher possibility of hydrogen bond formation. To further examine the mechanisms at the microscale, we considered three different models: a non-conjugate segment located at the middle, one-third, and the end of the polymer chain. Here, we consider that the DCPs are composed mainly of these three configurations with different weight percentages. Figure 6b shows the average number of hydrogen bond formations for different DCPs. The

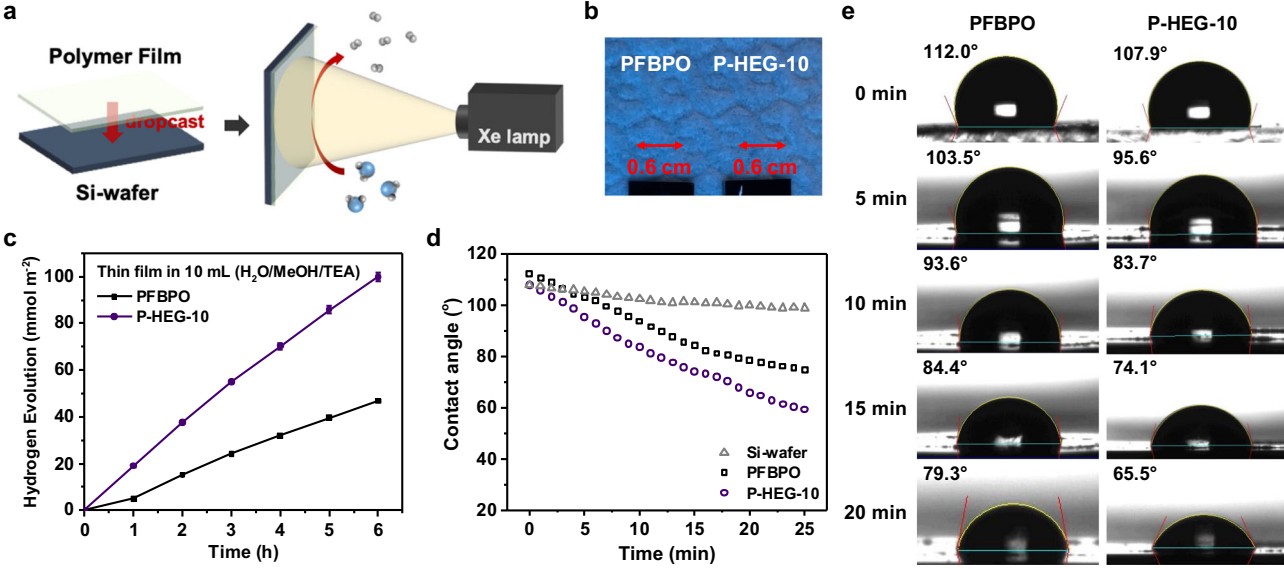

**Fig. 5 | Photocatalytic hydrogen evolution experiments in film. a** Schematic illustration of the photocatalytic hydrogen evolution in film systems. **b** Images of the 0.6 × 0.6 cm² film over PFBPO and P-HEG-10. **c** Time-dependent HER of PFBPO and P-HEG-10 film immersed in 10 mL of a H₂O/MeOH/TEA solution under 380–780 nm irradiation. **d** Correlation between the time and the water contact angles of Si-wafer, PFBPO, and P-HEG-10. **e** Time-dependence water contact angles without light illumination of PFBPO and P-HEG-10 films.

statistic of hydrogen bounds suggest that the system composed of 80% of the DCP with non-conjugated segments located at the center, 15% at 1/3 of the overall chain, and 5% at the end, in agreement with the trend of experimental findings. Figure 6c plots the radial distribution function and the morphology of each DCP chain. One can observe that DCP with 10 mol% HEG-modified exhibits the highest possibility of hydrogen bond formation and the probability of radial distribution function. In contrast, a 10 mol% EA-modified DCP shows a relatively low possibility of hydrogen bond formation. Simulation results also reveal that 10 mol% HEG-modified and 5 mol% EA-modified DCPs were entangled more than other DCP systems where other DCPs remained in single-chain formation.

## Discussion

In this study, we successfully developed a series of DCPs by inserting hydrophilic non-conjugated segments into the main-chain of hydrophobic CPs to produce photocatalysts with high HERs under visible-light irradiation. The DCPs with interrupted conjugation showed enhanced HER values in both the film state and the solution state compared to those of their hydrophobic photocatalyst counterparts under otherwise identical conditions. The hydrophilic non-conjugated segments effectively brought water into the inner polymer chain of main-chain-engineered DCPs, which increased the HER without obviously changing the semiconducting properties of the polymers, thereby overcoming a major limitation in the field. DCPs with hydrophilic segments of 5–10 mol% exhibited HERs comparable to those of hydrophobic CPs with the use of 33 vol.% organic solvent. Furthermore, in this study, we exploited a full atomistic study using molecular dynamics simulation to elucidate the interaction between water and DCPs which agrees well with the experimental measurements. This indicates that main-chain engineering using hydrophilic non-conjugated segments increases the possibility of water-DCP interaction and is more efficient in comparison with ordinary conjugate polymers. Importantly, we also demonstrated that our proposed approach is a universal route for synthesizing classes of DCPs with enhanced photocatalytic hydrogen evolution. The use of hydrophilic non-conjugated segments in the main-chain engineering of semiconducting polymers could lead to further optimization and greater molecular-design possibilities for developing high-performance

photocatalysts for the generation of clean and renewable energy and future industrial applications.

## Methods

### Synthesis of P-EG-5 photocatalyst

In all, 40 mL toluene and 10 mL water were injected into a sealed tube charged with the co-monomers F-B (643 mg 1.0 mmol), co-monomer BPO-Br (412 mg, 0.95 mmol), co-monomer EG-Br (19.0 mg, 0.05 mmol), Na₂CO₃ (1060 mg, 10.0 mmol), tetra-*n*-butylammonium bromide (16.0 mg, 0.05 mmol), and Pd(PPh₃)₄ (58.0 mg, 0.05 mmol). The mixture was degassed by bubbling with nitrogen for 30 min and then heated at 120 °C for 48 h. After reaction, the mixture was cooled to 25 °C and poured in MeOH. The precipitate was collected using membrane filtration. Purification of the polymer was performed through Soxhlet extraction with MeOH and hexane. Finally, the polymer was dissolved in hot CHCl₃, concentrated, and then precipitated in MeOH. The P-EG-5 was isolated as yellow-green powders in 80% yield. GPC (THF): $M_n$ 13.0 kg mol⁻¹; ¹H NMR (500 MHz, CDCl₃): δ 8.05 (d, $J = 10.5$ Hz), 7.93 (dd, $J = 19$ Hz, $J = 7.5$ Hz), 7.75-7.81 (m), 7.66 (d, $J = 7.5$ Hz), 7.55-7.60 (m), 7.49-7.52 (m), 7.43-7.47 (m), 4.39-4.42 (br), 1.97-2.03 (br), 1.13 (t, $J = 6.5$ Hz), 1.04-1.12 (br), 0.73-0.78 (m), 0.64 (br); ³¹P NMR (500 MHz, CDCl₃): δ 33.85.

### Synthesis of P-TEG-5 photocatalysts

In all, 40 mL toluene and 10 mL water were injected into a sealed tube charged with the co-monomer F-B (643 mg, 1.0 mmol), co-monomer BPO-Br (412 mg, 0.95 mmol), co-monomer TEG-Br (23.0 mg, 0.05 mmol), Na₂CO₃ (1060 mg, 10.0 mmol), TBAB (16.0 mg, 0.05 mmol), and Pd(PPh₃)₄ (58.0 mg, 0.05 mmol). The mixture was degassed by bubbling with nitrogen for 30 min and then heated at 120 °C for 48 h. After reaction, the mixture was cooled to 25 °C and poured in MeOH. The precipitate was collected using membrane filtration. Purification of the polymer was performed through Soxhlet extraction with MeOH and hexane. Finally, the polymer was dissolved in hot CHCl₃, concentrated, and then precipitated in MeOH. The P-TEG-5 was isolated as yellow-green powders in 76% yield. GPC (THF): $M_n$ 12.8 kg mol⁻¹; ¹H NMR (500 MHz, CDCl₃): δ 8.05 (d, $J = 10.5$ Hz), 7.93 (dd, $J = 19$ Hz, $J = 7.5$ Hz), 7.75-.81 (m), 7.66 (d, $J = 7.5$ Hz), 7.55-7.60 (m), 7.49-7.52 (m), 7.43-7.47 (m), 4.16 (br), 3.87 (br), 1.97-2.03 (br), 1.13 (t,

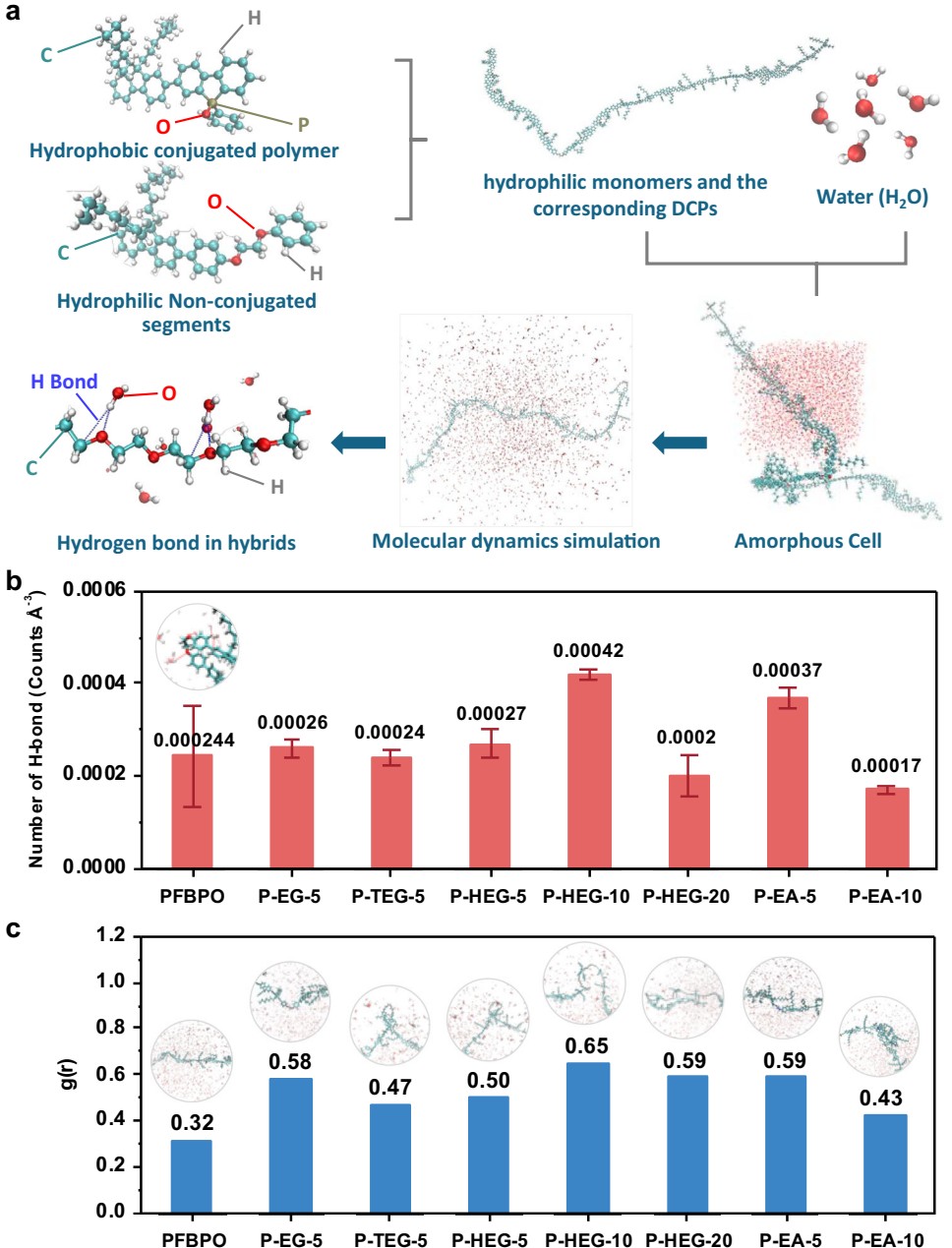

**Fig. 6 | Molecular dynamics simulations. a** The workflow of molecular dynamics study. The conjugate polymer models were built according to the overall mass and different repetition units. The system was filled with water in an amorphous cell to evaluate the hydro-bonding. **b** The statistics of hydrogen bounds and **c** the possibility of hydrogen bond formation of polymer photocatalysts.

$J = 6.5$ Hz), 1.04-1.12 (br), 0.73-0.78 (m), 0.64 (br); $^{31}$P NMR (500 MHz, CDCl$_3$): $\delta$ 33.87.

## Synthesis of P-HEG-5 photocatalysts

In all, 40 mL toluene and 10 mL water were injected into a sealed tube charged with the co-monomer F-B (643 mg 1.0 mmol), co-monomer BPO-Br (412 mg, 0.95 mmol), co-monomer HEG-Br (30.0 mg, 0.05 mmol), Na$_2$CO$_3$ (1060 mg, 10.0 mmol), TBAB (16.0 mg, 0.05 mmol), and Pd(PPh$_3$)$_4$ (58.0 mg, 0.05 mmol). The mixture was degassed by bubbling with nitrogen for 30 min and then heated at 120 °C for 48 h. After reaction, the mixture was cooled to 25 °C and poured in MeOH. The precipitate was collected using membrane filtration. Purification of the polymer was performed through Soxhlet extraction with MeOH and hexane. Finally, the polymer was dissolved in hot CHCl$_3$, concentrated, and then precipitated in MeOH. The P-

HEG-5 was isolated as yellow-green powders in 73% yield. GPC (THF): $M_n$ 14.1 kg mol$^{-1}$; $^1$H NMR (500 MHz, CDCl$_3$): $\delta$ 8.05 (d, $J = 10.5$ Hz), 7.93 (dd, $J = 19$ Hz, $J = 7.5$ Hz), 7.75-7.81 (m), 7.66 (d, $J = 7.5$ Hz), 7.55-7.60 (m), 7.49-7.52 (m), 7.43-7.47 (m), 4.13 (br), 3.83 (br), 3.62-3.64 (m), 1.97-2.03 (br), 1.13 (t, $J = 6.5$ Hz), 1.04-1.12 (br), 0.73-0.78 (m), 0.64 (br); $^{31}$P NMR (500 MHz, CDCl$_3$): $\delta$ 33.85.

## Synthesis of P-HEG-10 photocatalysts

In all, 40 mL toluene and 10 mL water were injected into a sealed tube charged with the co-monomer F-B (643 mg 1.0 mmol), co-monomer BPO-Br (390 mg, 0.90 mmol), co-monomer HEG-Br (59.0 mg, 0.10 mmol), Na$_2$CO$_3$ (1060 mg, 10.0 mmol), TBAB (16.0 mg, 0.05 mmol), and Pd(PPh$_3$)$_4$ (58.0 mg, 0.05 mmol). The mixture was degassed by bubbling with nitrogen for 30 min and then heated at 120 °C for 48 h. After reaction, the mixture was cooled to 25 °C and

poured in MeOH. The precipitate was collected using membrane filtration. Purification of the polymer was performed through Soxhlet extraction with MeOH and hexane. Finally, the polymer was dissolved in hot $CHCl_3$, concentrated, and then precipitated in MeOH. The P-HEG-10 was isolated as yellow-green powders in 71% yield. GPC (THF): $M_n$ 13.1 kg mol$^{-1}$; $^1$H NMR (500 MHz, CDCl$_3$): $\delta$ 8.05 (d, $J$ = 10.5 Hz), 7.93 (dd, $J$ = 19 Hz, $J$ = 7.5 Hz), 7.75-7.81 (m), 7.66 (d, $J$ = 7.5 Hz), 7.55-7.60 (m), 7.49-7.52 (m), 7.43-7.47 (m), 4.16 (br), 3.86 (br), 3.63-3.67 (m), 1.97-2.03 (br), 1.13 (t, $J$ = 6.5 Hz), 1.04-1.12 (br), 0.73-0.78 (m), 0.64 (br); $^{31}$P NMR (500 MHz, CDCl$_3$): $\delta$ 33.93.

### Synthesis of P-HEG-20 photocatalysts

In all, 40 mL toluene and 10 mL water were injected into a sealed tube charged with the co-monomer F-B (643 mg 1.0 mmol), co-monomer BPO-Br (347 mg, 0.80 mmol), co-monomer HEG-Br (118 mg, 0.20 mmol), Na$_2$CO$_3$ (1060 mg, 10.0 mmol), TBAB (16.0 mg, 0.05 mmol), and Pd(PPh$_3$)$_4$ (58.0 mg, 0.05 mmol). The mixture was degassed by bubbling with nitrogen for 30 min and then heated at 120 °C for 48 h. After reaction, the mixture was cooled to 25 °C and poured in MeOH. The precipitate was collected using membrane filtration. Purification of the polymer was performed through Soxhlet extraction with MeOH and hexane. Finally, the polymer was dissolved in hot $CHCl_3$, concentrated, and then precipitated in MeOH. The P-HEG-20 was isolated as yellow-green powders in 70% yield. GPC (THF): $M_n$ 13.4 kg mol$^{-1}$; $^1$H NMR (500 MHz, CDCl$_3$): $\delta$ 8.05 (d, $J$ = 10.5 Hz), 7.93 (dd, $J$ = 19 Hz, $J$ = 7.5 Hz), 7.75-7.81 (m), 7.66 (d, $J$ = 7.5 Hz), 7.55-7.60 (m), 7.49-7.52 (m), 7.43-7.47 (m), 4.16 (br), 3.86 (br), 3.63-3.67 (m),1.97-2.03 (br), 1.13 (t, $J$ = 6.5 Hz), 1.04-1.12 (br), 0.73-0.78 (m), 0.64 (br); $^{31}$P NMR (500 MHz, CDCl$_3$): $\delta$ 33.86.

### Synthesis of P-EA-5 photocatalysts

In all, 40 mL toluene and 10 mL water were injected into a sealed tube charged with the co-monomer F-B (643 mg 1.0 mmol), co-monomer BPO-Br (412 mg, 0.95 mmol), co-monomer EA-Br (19.0 mg, 0.05 mmol), Na$_2$CO$_3$ (1060 mg, 10.0 mmol), TBAB (16.0 mg, 0.05 mmol), and Pd(PPh$_3$)$_4$ (58.0 mg, 0.05 mmol). The mixture was degassed by bubbling with nitrogen for 30 min and then heated at 120 °C for 48 h. After reaction, the mixture was cooled to 25 °C and poured in MeOH. The precipitate was collected using membrane filtration. Purification of the polymer was performed through Soxhlet extraction with MeOH and hexane. Finally, the polymer was dissolved in hot $CHCl_3$, concentrated, and then precipitated in MeOH. The P-EA-5 was isolated as yellow-green powders in 79% yield. GPC (THF): $M_n$ 15.2 kg mol$^{-1}$; $^1$H NMR (500 MHz, CDCl$_3$): $\delta$ 8.05 (d, $J$ = 10.5 Hz), 7.93 (dd, $J$ = 19 Hz, $J$ = 7.5 Hz), 7.75-7.81 (m), 7.66 (d, $J$ = 7.5 Hz), 7.55-7.60 (m), 7.49-7.52 (m), 7.43-7.47 (m), 3.55 (m), 1.97-2.03 (br), 1.13 (t, $J$ = 6.5 Hz), 1.04-1.12 (br), 0.73-0.78 (m), 0.64 (br); $^{31}$P NMR (500 MHz, CDCl$_3$): $\delta$ 33.88.

### Synthesis of P-EA-10 photocatalysts

In all, 40 mL toluene and 10 mL water were injected into a sealed tube charged with the co-monomer F-B (643 mg 1.0 mmol), co-monomer BPO-Br (390 mg, 0.90 mmol), co-monomer EA-Br (37.0 mg, 0.10 mmol), Na$_2$CO$_3$ (1060 mg, 10.0 mmol), TBAB (16.0 mg, 0.05 mmol), and Pd(PPh$_3$)$_4$ (58.0 mg, 0.05 mmol). The mixture was degassed by bubbling with nitrogen for 30 min and then heated at 120 °C for 48 h. After reaction, the mixture was cooled to 25 °C and poured in MeOH. The precipitate was collected using membrane filtration. Purification of the polymer was performed through Soxhlet extraction with MeOH and hexane. Finally, the polymer was dissolved in hot $CHCl_3$, concentrated, and then precipitated in MeOH. The P-EA-10 was isolated as yellow-green powders in 75% yield. GPC (THF): $M_n$ 13.9 kg mol$^{-1}$; $^1$H NMR (500 MHz, CDCl$_3$): $\delta$ 8.05 (d, $J$ = 10.5 Hz), 7.93 (dd, $J$ = 19 Hz, $J$ = 7.5 Hz), 7.75-7.81 (m), 7.66 (d, $J$ = 7.5 Hz), 7.49-7.52 (m), 7.43-7.47 (m), 3.56-3.60 (m), 1.97-2.03 (br), 1.13 (t,

$J$ = 6.5 Hz), 1.04-1.12 (br), 0.73-0.78 (m), 0.64 (br); $^{31}$P NMR (500 MHz, CDCl$_3$): $\delta$ 33.87.

### Photocatalytic hydrogen evolution

Hydrogen was detected using a Shimadzu GC-2014 gas chromatograph equipped with a thermal conductivity detector, with Ar as carrier gas. In a typical measurement, 5 mg of polymer powder and 10 mL of a mixed solution consisting of 33.3 vol.% H$_2$O, 33.3 vol.% MeOH, and 33.3 vol.% TEA are added to the cuvette and sealed with a septum. The resulting mixture was degassed by bubbling with argon gas for 10 min, prior to illumination. The suspension was illuminated with a 350-W Xe-lamp (1000 W/m$^2$, $\lambda$ > 420 nm), maintaining its temperature at 27 ± 1 °C under atmospheric pressure and keeping the distance between the reaction mixture and light source fixed.

### Photocurrent measurements

A Zahner Zennium E works station equipped with three-electrode cell consists of Ag/AgCl as reference electrode (3 M NaCl), Pt wire as counter electrode, and fluorine doped tin oxide (FTO) glass as working electrode. The polymer sample (5 mg) was dispersed in dry toluene solution (1 mL) and ultra-sonicated for 1 h. Then, 200 μL of the as-prepared polymer suspension was spin-coated on FTO-glass with an active area of 1.0 cm$^2$. The electrolyte was an aqueous solution of 0.5 M Na$_2$SO$_4$. We measured the photocurrent generated using LED irradiation (constant potential = 1.5 V) with the light switched on-off with intervals of 20 s.

### Molecular dynamics

Molecular simulations were performed by Materials Studio. An overall workflow is depicted in Fig. 6a. We adopted the Dreiding force field as the interatomic potential, and built a single chain model in the center of the periodic box. Before dynamic simulations, geometry optimization was performed to relax the artificial polymer chain to obtain a suitable initial model. Then, the NVT ensemble was chosen to run the dynamic simulations at a temperature of 298 K in 1 ns for relaxing the polymer chain within the water. Finally, the NPT ensemble was adopted to drive with a temperature and pressure of 298 K and 1 atm, respectively, for 1 ns to obtain an equilibrium structure. After dynamic simulations, the trajectory file was analyzed to obtain the radial distribution function to predict the possibility of hydrogen bond formation in Fig. 6b, and the number of hydrogen bonds over unit volume was also statistical, as shown in Fig. 6a.

## Data availability

The data that support the findings within this paper are available within the article and the Supplementary Information file, or available from the corresponding author upon request. Source data are provided with this paper.

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

## Acknowledgements
The authors gratefully acknowledge the financial support of the National Science and Technology Council of Taiwan (NSTC 111-2221-E-007-004-; NSTC 111-2628-E-007-009-; and NSTC 110-2622-8-007-011-). The authors would like to thank the Instrumentation Center at National Tsing Hua University for providing the MS, NMR, and EPR measurement. The authors would like to thank the Center for Nanotechnology, Materials Science, and Microsystems at National Tsing Hua University for providing the DB-FIB measurement. The National Synchrotron Radiation Research Centre (NSRRC), Taiwan, is gratefully acknowledged for SXAS/WXAS measurements. The computing resources were supported by TAIWANIA at the National Center for High-Performance Computing (NCHC) in Taiwan.

## Author contributions
C.-L.C. designed, planned, and performed the experiments and wrote the manuscript. W.-C.L., L.-Y.T., and T.-F.H. prepared the polymer photocatalysts. J.J. and Y.-J.L. carried out the transient experiments. C.-H.S., C.-W.C., and C.-H.Y. performed the MD calculations. S.-Y.C., H.T., and T.M. assisted in the experiments and characterizations. W.-Y.J., C.-W.T., C.-C.H., and A.M.E. performed the electrochemical experiments. C.-Y.C. performed the SAXS/WAXS experiments. H.-H.C. supervised the work on polymer synthesis and photocatalysis. All the authors participated in the interpretation and discussion of the results.

## Competing interests
The authors declare no competing interests.

## Additional information

¹Department of Chemical Engineering, National Tsing Hua University, Hsinchu 300044, Taiwan. ²Research Center for Applied Sciences, Academia Sinica,
Taipei 115024, Taiwan. ³Department of Engineering Science, National Cheng Kung University, Tainan 701401, Taiwan. ⁴Energy Catalyst Technology Group,
Energy Process Research Institute, National Institute of Advanced Industrial Science and Technology, Ibaraki 305-8559, Japan. ⁵Department of Chemical
Engineering, National Chung Hsing University, Taichung 402202, Taiwan. ⁶Department of Molecular Science and Engineering, National Taipei University of
Technology, Taipei 106344, Taiwan. ✉e-mail: hhchou@mx.nthu.edu.tw

