## [Peer Review File · Nature Communications]

Title: Main-chain engineering of polymer photocatalysts with hydrophilic non-conjugated segments for visible-light-driven hydrogen evolutionREVIEWER COMMENTS

Reviewer #1 (Remarks to the Author):

Investigation of organic polymeric photocatalysts for hydrogen production is a timely research topic. Many strategies have been reported in this research field to improve activity of the photocatalysts. Modifying polymer structures is one of them. In this work, Chan et al., put hydrophilic chains in the main-bone of the polymers to increase the interaction between protons and the polymers, showing improved hydrogen production activity. However, the absorption of the reported polymers is unsatisfactory, most of the absorption is located at UV and near UV-region. If the authors tried to develop a new method, rather than new polymers, then I can not judge how the method is generalized in other polymers.

Some other comments:

1. Why did the authors use Tauc plot to get the bandgap of organic polymers, but at same time use HOMO and LUMO levels?
2. Fig 3 caption is completely not informative. All key information for each sub-figure is missing.
3. Did the pH keep consistent when the co-catalyst Pt was added in Fig 3a experiments?
4. In Figure 3b, no significant difference in lifetimes of polymers is actually observed.
5. In Figure 3e, did or how did the author control the thickness of all polymer films same to make sure the comparison of photocurrent is valid?
6. In Figure 3f, Why does not the reduction potential of P-10HEG ca. -0.7 vs. Ag/AgCl match with the LUMO level of the polymer indicated in Figure 2, -3.16 eV vs. vacuum?
7. The authors synthesized a side-chain polymers PFTEGBPO to compare with P-10HEG to show their method of putting hydrophilic chain the polymer bone is valid. However, from the photos they provided in Figure 42, PFTEGBPO is more soluble in the photocatalytic solution, meaning PFTEGBPO interacts with protons more than P-10HEG does. This is not consistent with what the authors claimed thoroughly in this work.

Reviewer #2 (Remarks to the Author):

This is an interesting manuscript containing interesting data. The idea to introduce hydrophilic monomers in the polymer backbone for photocatalytic hydrogen evolution is novel as far as I am aware. There are, however, some issues that I think need to be addressed before publication, see below, and I think the authors slightly overhype their results, see talk of record values.

1. It is confusing that the authors label their materials as P5, P10 and P20, as other authors have already used the same label for other polymers. The confusion is worst in the case of P10, poly(dibenzosulfone), one of the more active materials reported in the literature. For example, supplementary table 1 in the supporting information contains two P10s, the original P10 from the literature and the P10-HEG

reported here, and the materials are quite different in structure.

2. The authors refer to the measured AQY value of P-10HEG at 460 nm, 17.83%, as record high in the abstract. This raises several questions. Firstly, the AQY reported here for P10-HEG at 420 nm, 18.19% is actually higher, and secondly, as can be seen from supplementary table 1 in the supporting information other polymers, e.g., P64, have even higher AQY values at 420 nm. This statement only feels justifiable in the sense that 17.83 might be highest value reported at 460 nm but that it's probably simply because few papers report AQY values at 460 nm.

3. Fig. 1 in the main text and Supplementary scheme 1 seem to imply that the fluorene unit is always connected to two 5-phenyldibenzophosphole 5-oxide units. However, that seems naively at odds with the synthesis chemistry, where if I'm not mistaken the fluorene can be either coupled with 5-phenyldibenzophosphole 5-oxide units or one of the hydrophilic non-conjugated building blocks. If so, the structure is more complicated than suggested.

4. Following on from the above, if I'm not mistaken these are statistical co-polymers, in the sense that the different monomers are not ordered, other than the fluorene unit can never be linked to another fluorene unit, and that there will be a distribution of conjugated and hydrophilic blocks of different lengths. If so, the authors might want to clarify this to the reader and avoid the term block as that would suggest that these are blockpolymers, which these are not.

5. I am confused by the statement "the differences between the HOMO/LUMO levels and the optical E_g were small, indicating that hydrophilic non-conjugated building blocks could be molecularly engineered into the main-chain of the target CPs using the present approach without significantly affecting the physicochemical and optical properties". If I'm not mistaken the authors use the E_g value to calculate the LUMO from the measured HOMO level and hence the difference between the LUMO and HOMO levels should be exactly E_g but perhaps I am missing something.

6. One should be slightly careful with the HOMO and LUMO levels measured by PESA shown in Fig. 2C. Not only are they not truly HOMO and LUMO values but rather more like adiabatic ionisation potentials and electron affinity values (the HOMO/LUMO terminology is problematic when the material has time to relax on the timescale of the photoelectron experiment) but the PESA is done in air while the hydrogen evolution experiments are done in water. When immersing the polymer in water the adiabatic ionisation potentials and electron affinity values will move closer together as water stabilises the charges in the polymer. As a result, a material typically has a smaller thermodynamic driving force for reduction and oxidation than suggested by plotting the PESA derived data versus the half-reaction potentials, as done in Fig. 2C.

7. What is responsible for the change in contact angle under illumination?

8. I don't think the authors specify the relative proportions of water, TEA and methanol in their reaction mixture for the hydrogen evolution. Is it 1:1:1 on volume basis? Similar what was the ratio between

water and TEA in the methanol free experiments (or the TEA concentration)? The reason methanol is typically used is because water and TEA don't mix well at high TEA concentrations and phase separates.

9. I am slightly not impressed with the fact that only hydrogen evolution data up to 6 hours is shown. For such a paper, I would have expected to see data for at least the best performing material of 24 hours or more. Other authors report data for 48 hours or more.

10. The discussion of the EPR data on page 8 and how it's shown in Supplementary Fig. 27 is potentially confusing. The way I read the data and discussion is that the materials already show an EPR signal in the dark, which is surely not correct. Also, was the EPR done on the dry polymer or the polymer immersed in water and/or the water/methanol/TEA mixture?

11. Does Supplementary Fig. 41 show the influence of the Pd co-catalyst on the HER of P-10HEG and PFBPO as the caption suggests or the effect of adding Pt instead?

12. The authors should really share the MD structures used to calculate the data in Fig. 6b and 6C as additional electronic supporting material.

Reviewer #3 (Remarks to the Author):

In this work, the authors reported a strategy for improving the photocatalytic hydrogen production activity by introducing hydrophilic non-conjugated building blocks in polymer main-chain and developed a series of organic photocatalysts. A moderate hydrogen evolution rate of 16.8 mmol m⁻² h⁻¹ under visible-light irradiation was obtained by P-10HEG with interrupted conjugation. This work is interesting in the field of photocatalytic hydrogen production by polymer-based photocatalysts. Therefore, this manuscript might be suitable for the publication in Nat. Commun. after the following issues are addressed.

1. P-10HEG with the longest fluorescence lifetimes has the highest photocatalytic hydrogen evolution rate. It seems that the longer fluorescence lifetimes of the photocatalysts, the higher photocatalytic hydrogen evolution activity. Why does the organic photocatalyst of P-5HEG show a higher hydrogen evolution rate (5.38 ± 0.31 mmol h⁻¹ g⁻¹) than P-20HEG (3.74 ± 0.36 mmol h⁻¹ g⁻¹), while they have the same fluorescence lifetime (1.11 ns)? Is there a definite relationship between the photocatalytic hydrogen evolution activity and fluorescence lifetime for organic photocatalysts? If not, it is not suitable to evaluate the photocatalytic activity by time-resolved photoluminescence spectroscopy.
2. The authors mentioned that the type and content of hydrophilic non-conjugated building blocks in the hydrophobic CPs do not significantly affect the hydrophilicity of the DCP films due to packing of the polymers. How about the dispersion ability of the polymers in pure water? A dynamic light scattering (DLS) test of the polymers in pure water is suggested to be provided.
3. The introduction of hydrophilic chain segment leads to a swelling for the polymer thin-film of P-10HEG after contacting with water, which might be disadvantageous in the photocatalytic stability of

polymer thin-film for P-10HEG.

4. I have noticed that the samples of hydrophilic P-20HEG and P-10EA show a lower HER than the hydrophobic PFBPO. This should be discussed in detail.

Reviewer #4 (Remarks to the Author):

Polymer photocatalysts are an interesting alternative to the more commonly studied inorganic materials for water splitting/hydrogen evolution. A major issue with most polymer photocatalysts is that water has a low level of access to the polymer with large aggregates forming. This is arguably the critical issue in polymer photocatalysis to solve as the water (or electron donor)/polymer interface is likely to be the site of charge separation so minimising exciton transport distances is critical. Furthermore the water can play an important role in stabilising polymer polaron states. The state of the art is to include hydrophilic groups on conjugated polymers. Here the authors report a really interesting and novel approach which is to disrupt the conjugated structure and put hydrophilic regions into the polymer backbone that break the conjugation.

The concept will be interesting to the community and I encourage its exploration. But the current level experimental evidence linking the levels of photocatalytic activity to the level of polymer hydration/swelling is insufficient to prove the claims. Furthermore, the changes in photocatalytic activity are relatively small (largest increase is 50%).

Main points:

1. No change in contact angle for water is seen on the polymer films upon inclusion of the hydrophilic groups and instead the authors rely on the particle size as determined by DLS to assess polymer swelling. The particle size in suspension is dependent upon many factors and not just the degree of hydration. To evaluate the papers central hypothesis a more robust measure of polymer hydration is needed.
2. There is no correlation between particle sizes and photocatalytic activity. For example P-20HEG has the largest particle size but a photocatalytic activity that is less than the control material. This is perhaps not surprising as noted above, the DLS is not a good measure of hydration.
3. Data is presented for only a small selection of the materials for several of the important experiments. For the EPR and PL quenching data for P-5EG, P10-HEG and P-5EA is provided but it is not clear why these samples are chosen as they do not allow for assessment of percentage of hydrophilic group on behaviour or comparison between the hydrophilic groups. For the CV only P-10HEG is shown.
4. The CV of P0-HEG has a redox couple at ~ -0.7 V Ag/AgCl. This is -0.5 Vs NHE. Assuming the redox couple is pH independent this means the driving force for H₂ evolution is ~ 250 meV at pH 12.7, significantly less than Fig 2 indicates. This is perhaps ok as the XPS and optical gap is reporting on the vertical transition and the CV on the potential of polymer reduction. But it would be useful to show the

CV's for the other polymers and also to check the pH dependence of the redox couple.

5. UV/Vis spectra show that several of the polymers have a broad feature at long-wavelengths which is not described. Fig 4d shows it is photo-catalytically active transition so it is important to discuss.

6. Fig 3 b shows the lifetimes of the PL are essentially the same but table 1 lists quite large differences in the lifetime (but no errors). All values look like they are very close to the instrument response function. I caution against over interpretation of these.

7. Page 8, line 163. EIS shows a lower resistance which indicates enhanced charge transport. Why would increasing the water/polymer interface area improve charge transport? I expect the primary role of the water is to increase charge separation yields. I would have expected disruption of the conjugation would decrease charge transport.

Reviewer #5 (Remarks to the Author):

In this paper, the authors report a new type of conjugated polymer for use in photocatalysis. The authors provide characterization of the materials, as well as their photocatalytic activity and ultimately, conclude that the amount of hydrogen bonding of the system with water dictates the efficacy towards water splitting. While the work is interesting, it is not immediately apparent to me that the work provides new insight in terms of using conjugated polymers for water splitting for reasons primarily outlined in point 1 below. There is also a little bit of data missing which makes it difficult to judge some of the claims made in the paper.

1. The authors indicate that this is the first time that a system with broken conjugation was shown to be effective in water splitting. However, I disagree with this concept because Figure 2 shows that all the UV-vis spectra are similar. This similarity indicates that the effective conjugation length of all systems is similar, which in turn means that the conjugation was not broken or that the parent PFBPO has broken conjugation in the first place. In that sense, this claim is misleading.

2. For the EIS data shown on Page 9, the authors should indicate what equivalent circuit was used to calculate the resistance.

3. Page 8: "The EIS data recorded in the dark showed a semicircular curve (Fig. 3d), where the DCPs had resistance values lower than that of PFBPO due to their enhanced charge transport" The data, as shown, is inconclusive to support that the DCPs had enhanced charge transport. Related to point 2 above, more discussion needs to be provided as to how the authors determined if the arc shown in the Nyquist plot is DCP resistance, not electrolyte or interface resistance.

4. For Figure S38, please indicate in the figure caption what solvent was used for the DLS measurement.

5. Pages 9-10: The DLS results are discussed here and it is postulated that the DCPs swelled in an aqueous solution. Can the authors also provide the DLS data in a non-aqueous solution too to support that the bigger size is indeed due to swelling instead of being an intrinsic property of the particles?

National Tsing Hua University

Department of Chemical Engineering

No. 101, Sec. 2, Kuang-Fu Road
Hsinchu 300044, Taiwan

Responses to Reviewer #1

Comments to the Author

Comments:

Investigation of organic polymeric photocatalysts for hydrogen production is a timely research topic. Many strategies have been reported in this research field to improve activity of the photocatalysts. Modifying polymer structures is one of them. In this work, Chan et al., put hydrophilic chains in the main-bone of the polymers to increase the interaction between protons and the polymers, showing improved hydrogen production activity. However, the absorption of the reported polymers is unsatisfactory, most of the absorption is located at UV and near UV-region. If the authors tried to develop a new method, rather than new polymers, then I can not judge how the method is generalized in other polymers.

The manuscript was carefully revised based on the suggestions and comments, and we hope that the revised manuscript is suitable for publication in this esteemed journal.

Thank you very much for the reviewer's comments. In our original manuscript, actually, in addition to applying the method to PFBPO, we did also apply this method to another well-known polymer, PF8BT (where the threshold point of absorption band is around 500 nm, see Supporting Information, Fig. 51), to demonstrate our method is universal and is not only for the absorption of polymers located at UV or near UV-region. Therefore, this demonstrates that we can easily applied our method to the other polymer, which the absorption is located at visible-light region.

Moreover, in order to further demonstrate the universality of our method again, we further applied this method to one more polymer, PCPDTBSO, in the revised manuscript. It can be seen that the absorption maxima of these three polymers are red-shifted from 400 nm for P-HEG-10 to 450 nm for PF8BT-HEG-10 and then to 520 nm for PCPDTBSO-HEG-10, and the enhanced photocatalytic hydrogen evolution results are obtained in all three polymers. All these demonstrate that our method is new and is universal.

The main-chain engineering approach was applied on polymers with different absorption maxima, PFBPO ($\lambda_{\text{max.}} = 400$ nm), PF8BT ($\lambda_{\text{max.}} = 450$ nm) and PCPDTBSO ($\lambda_{\text{max.}} = 520$ nm), to demonstrate that our approach is novel and universal.

Comment 1. Why did the authors use Tauc plot to get the bandgap of organic polymers, but at same time use HOMO and LUMO levels?

Response: We thank reviewer for pointing this out. As mentioned in our manuscript on page 6, we measured the HOMO levels of the prepared polymers using a photoelectronic spectrometer. Then, we calculated the LUMO levels using the equation of $E_{\text{LUMO}} = E_{\text{HOMO}} + E_{\text{g}}$, where the E_{g} (bandgap) values of the polymers calculated from Tauc plots. This method was considered a standard approach for getting the bandgap of organic polymers, and has been reported in many previous reports (*J. Am. Chem. Soc.* **139**, 5216 (2017); *J. Am. Chem. Soc.* **140**, 1423 (2018); *Chem* **5**, 1632 (2019); *Commun. Mater.* **1**, 90 (2020); *J. Am. Chem. Soc.* **142**, 20763 (2020); *Angew. Chem. Int. Ed.* **59**, 16902 (2020); *Adv. Mater.* **33**, 2006274 (2021); *Nat. Commun.* **12**, 483 (2021); *ACS Catal.* **11**, 13266 (2021); *Angew. Chem. Int. Ed.* **60**, 2 (2021); *Appl. Catal. B* **307**, 121144 (2022)).

Comment 2. Fig 3 caption is completely not informative. All key information for each sub-figure is missing.

Response: Thank you for your valuable comments. Per the reviewer's suggestion, we added all the key information for each sub-figure in the caption of Fig. 3.

As results, a new caption was added in the revised manuscript on page 9:

Fig. 3 Optical and electrochemical properties of polymers. **a** Photoluminescence spectra were obtained in suspensions (5 mg photocatalyst in 10 mL solution mixture consisting of equal volumes of H₂O, MeOH, and TEA). **b** Time-resolved photoluminescence spectra were obtained in suspensions (5 mg photocatalyst in 10 mL solution mixture consisting of equal volumes of H₂O, MeOH, and TEA). Samples were excited with a 405 nm laser and emission was measured at 455 nm. **c** EPR spectra were collected in the solid state under additional light illumination. **d** Electrochemical impedance spectra of polymers were carried out in dark with an AC potential frequency ranging from 0.1 Hz to 100 kHz. In the equivalent circuit, R_s represents the circuit series-resistance, R_{ct} is the charge transfer resistance across the interface, and C_{dl} is the capacitance phase element of the semiconductor-electrolyte interface. **e** Photocurrent were generated upon light on-off switching. **f** Wide-angle X-ray scattering profiles of PFBPO and P-HEG-10 were measured in a mixture solution consisting of equal volumes of H₂O, MeOH, and TEA.

Comment 3. Did the pH keep consistent when the co-catalyst Pt was added in Fig 3a experiments?

Response: We are grateful for the reviewer's suggestion. Per the reviewer's suggestion, we checked the pH and observed it did keep consistent after adding Pt co-catalyst in photoluminescence quenching experiments (Supplementary Fig. 28), in which the pH keeps at 12.4. We also added this information into the caption of Supplementary Fig. 28.

Supplementary Fig. 28 Photoluminescence spectra of **a** PFBPO, **b** P-EG-5, **c** P-TEG-5, **d** P-HEG-5, **e** P-HEG-10, **f** P-HEG-20, **g** P-EA-5, and **h** P-EA-10 were obtained in suspensions (5 mg photocatalyst in 10 mL mixture consisting of equal volumes of H₂O, MeOH, and TEA) before (solid line) and after (dash line) adding 3 wt.% H₂PtCl₆ co-catalyst, respectively, in which the pH keeps consistent at 12.4.

Comment 4. In Figure 3b, no significant difference in lifetimes of polymers is actually observed.

Response: Thank you for drawing our attention to this issue. Per the reviewer’s suggestion, we carefully remeasured time-resolved photoluminescence data and added the errors in Table 1. To avoid the misleading description of our data, per the reviewer’s suggestion, we modified the paragraph on time-resolved photoluminescence on page 7 of the revised manuscript.

“Time-resolved photoluminescence spectroscopy (Fig. 3b) shows that there is no significant difference in exciton lifetimes between all polymers, with their excited-state lifetimes ranging from 0.85–1.07 ns. To sum up, the HOMO/LUMO levels, optical E_g , photoluminescence spectra, and exciton lifetime of all polymers showed no obvious difference, indicating that hydrophilic non-conjugated segments could be molecularly engineered into the main-chain of the target CPs using the present approach without significantly affecting the physicochemical and optical properties.”

Fig. 3 Optical and electrochemical properties of polymers. ...b Time-resolved photoluminescence spectra were obtained in suspensions (5 mg photocatalyst in 10 mL mixture consisting of equal volumes of H₂O, MeOH, and TEA). Samples were excited with a 405 nm laser and emission was measured at 455 nm. **c...**

Comment 5. In Figure 3e, did or how did the author control the thickness of all polymer films same to make sure the comparison of photocurrent is valid?

Response: Thanks for your valuable comments. As the reviewer mentioned, the effect of thickness of the polymers on the photocurrent values is important. That is why in this study, we fabricated all polymer films under the same conditions by consisting the concentration, revolutions per minute, and spin-coating time, to obtain the comparable thickness of all polymer films to make a valid comparison. We further measured the thickness of polymer film using dual beam-focused ion beam (DB-FIB), and the thickness range was approximately 571–585 nm, demonstrating that all polymer films have comparable thickness to each other. We have added the DB-FIB images in Supplementary Fig. 32 of the revised manuscript.

Supplementary Fig. 32 DB-FIB images of all polymers used to measure polymer film thickness.

As results, a sentence was added in Page 9 of the revised manuscript.

“Transient photocurrent measurements used to investigate the photoresponse of all polymers. After fabricating the polymer films, we measured the thickness of polymer films using dual beam-focused ion beam (DB-FIB), and the thickness range of the polymer films was approximately 571–585 nm (Supplementary Fig. 32), demonstrating that all polymer films have comparable thickness to each other.”

Comment 6. In Figure 3f, why does not the reduction potential of P-10HEG ca. -0.7 vs. Ag/AgCl match with the LUMO level of the polymer indicated in Figure 2, -3.16 eV vs. vacuum?

Response: Thank you for drawing our attention to this issue. Per the reviewer’s suggestion, we have carefully rechecked and remeasured cyclic voltammetry (CV) spectroscopy. As result, the oxidation potential of these polymers measured using cyclic voltammetry (CV) exhibited comparable results to photoelectronic spectroscopy. We have added CV data in Supplementary Fig. 27 of the revised manuscript.

Polymer	Oxidation Potential (V vs NHE)	$E_{\text{HOMO,CV}}$ (eV)	$E_{\text{HOMO,PESA}}$ (eV)
PFBPO	1.27	-5.67	-5.82
P-EG-5	1.35	-5.74	-5.92
P-TEG-5	1.37	-5.77	-5.91
P-HEG-5	1.33	-5.73	-5.91
P-HEG-10	1.43	-5.83	-5.92
P-HEG-20	1.35	-5.75	-5.90
P-EA-5	1.34	-5.73	-5.93
P-EA-10	1.42	-5.82	-5.91

Supplementary Fig. 27 Cyclic voltammety measurements of **a** PFBPO, **b** P-EG-5, **c** P-TEG-5, **d** P-HEG-5, **e** P-HEG-10, **f** P-HEG-20, **g** P-EA-5, **h** P-EA-10, and **i** ferrocene/ferrocenium couple. Ferrocene/ferrocenium (Fc/Fc^+) redox potential was measured under the same condition to calibrate the reference electrode. The HOMO levels are determined as follows: $E_{\text{HOMO}} = -(E_{\text{ox, onset}} - E(\text{Fc}/\text{Fc}^+) + 4.8)$ eV.

As results, a sentence was added in Page 6 of the revised manuscript.

“Their highest occupied molecular orbital (HOMO) levels ranged from -5.90 to -5.93 eV, as measured using a photoelectronic spectrometer (Supplementary Fig. 26). Furthermore, the oxidation potential of these polymers measured using cyclic voltammety (CV) exhibited comparable results to

photoelectronic spectroscopy (Supplementary Fig. 27).”

Comment 7. The authors synthesized a side-chain polymers PFTEGBPO to compare with P-10HEG to show their method of putting hydrophilic chain the polymer bone is valid. However, from the photos they provided in Figure 42, PFTEGBPO is more soluble in the photocatalytic solution, meaning PFTEGBPO interacts with protons more than P-10HEG does. This is not consistent with what the authors claimed thoroughly in this work.

Response: Thank you for your valuable comments. Although hydrophilicity is one of the factors affecting the photocatalytic hydrogen evolution activity, it cannot be ignored the other factors, such as the light absorption of the photocatalytic solution. Previous study has proven that proper aggregation is beneficial to enhance photocatalytic HER, which is related to multiple scattering effects and light absorption [*J. Mater. Chem. A* **7**, 2490 (2019)].

To demonstrate this effect, we further measured and compared the transmission spectra of P-HEG-10 and PFTEGBPO under otherwise identical conditions (5 mg photocatalyst in 10 mL solution mixture consisting of equal volumes of H₂O, MeOH, and TEA). As shown in Supplementary Fig. 50, PFTEGBPO exhibited higher transmittance in the range of visible-light region due to its high solubility, resulting in the weaker light absorption and poor HER performance, compared to that of P-HEG-10.

Supplementary Fig. 50 Comparison of applicability of main-chain-engineered DCP and side-chain-engineered CP. **a** Time-dependent HER under standard condition (5 mg photocatalyst in 10 mL solution mixture consisting of equal volumes of H₂O, MeOH, and TEA). **b** Transmission spectra under otherwise identical conditions with hydrogen evolution experiments. **c** Molecular structure of side-chain-engineered PFTEGBPO. **d** Images of P-HEG-10 and PFTEGBPO photocatalytic solutions.

As results, a sentence was added in Page 15 of the revised manuscript.

“To further demonstrate the advantages and applicability of our design approach, we synthesized

a side-chain-engineered CP (denoted as PFTEGBPO) to compare its photocatalytic HER with that of the P-HEG-10 DCPs (Fig. 4f). We considered it is because PFTEGBPO exhibits higher transmittance in the range of 380–780 nm, as shown in Supplementary Fig. 50, resulting in the inefficient light harvesting and poor HER performance.”

Responses to Reviewer #2

Comments to the Author

Comments:

This is an interesting manuscript containing interesting data. The idea to introduce hydrophilic monomers in the polymer backbone for photocatalytic hydrogen evolution is novel as far as I am aware. There are, however, some issues that I think need to be addressed before publication, see below, and I think the authors slightly overhype their results, see talk of record values.

We thank the reviewer for your positive comments and point out the novelty of this work. The manuscript was carefully revised based on your suggestions and comments, and we hope that the revised manuscript is suitable for publication in this esteemed journal.

Comment 1. It is confusing that the authors label their materials as P5, P10 and P20, as other authors have already used the same label for other polymers. The confusion is worst in the case of P10, poly(dibenzosulfone), one of the more active materials reported in the literature. For example, supplementary table 1 in the supporting information contains two P10s, the original P10 from the literature and the P10-HEG reported here, and the materials are quite different in structure.

Response: We thank the reviewer for the suggestions. Per the reviewer’s suggestion, we have relabeled all polymers in this study to avoid confusing the readers. As results, our resulting polymers are labeled as P-EG-5, P-TEG-5, P-HEG-5, P-HEG-10, P-HEG-20, P-EA-5, and P-EA-10, respectively, in the revised manuscript.

Comment 2. The authors refer to the measured AQY value of P-10HEG at 460 nm, 17.83%, as record high in the abstract. This raises several questions. Firstly, the AQY reported here for P10-HEG at 420 nm, 18.19% is actually higher, and secondly, as can be seen from supplementary table 1 in the supporting information other polymers, e.g., P64, have even higher AQY values at 420 nm. This statement only feels justifiable in the sense that 17.83 might be highest value reported at 460 nm but that it’s probably simply because few papers report AQY values at 460 nm.

Response: Thank you for your valuable comments. In most of polymer photocatalysts, the AQY value decreases dramatically when the irradiation is at the longer single wavelength. In order to demonstrate the capability of our polymer at longer single wavelength (the single wavelength irradiation, which is > 450 nm), we measured the AQY of P-10HEG to prove its excellent photocatalytic performance at 460 nm.

Per the reviewer’s suggestion, to avoid the misleading description of our data, we modified the sentence in the abstract of revised manuscript on page 2.

“P-HEG-10 with 10 mol% ethylene glycol-based hydrophilic segments achieved an outstanding apparent quantum yield of 17.83% under 460 nm monochromatic light irradiation in solution and an

excellent hydrogen evolution rate of $16.8 \text{ mmol m}^{-2} \text{ h}^{-1}$ in thin-film.”

Comment 3. Fig. 1 in the main text and Supplementary scheme 1 seem to imply that the fluorene unit is always connected to two 5-phenyldibenzophosphole 5-oxide units. However, that seems naively at odds with the synthesis chemistry, where if I'm not mistaken the fluorene can be either coupled with 5-phenyldibenzophosphole 5-oxide units or one of the hydrophilic non-conjugated building blocks. If so, the structure is more complicated than suggested.

Response: We are grateful for the reviewer's suggestion. As mentioned by the reviewer, fluorene with boronic ester groups can be coupled with either 5-phenyldibenzophosphine 5-oxide units with bromine groups or one of the hydrophilic non-conjugated units with bromine groups. Furthermore, when we adjusted the molar ratio of 5-phenyldibenzophosphine 5-oxide units to hydrophilic non-conjugated units during polymerization, the moles of fluorene units remained consistent. Therefore, according to the reviewer's suggestion, we have redrawn the chemical structures of the polymers in Fig. 1 and Supplementary Scheme 1 of the revised manuscript.

Fig. 1 Schematic illustration and polymer structures. A Schematic illustration of the polymer photocatalysts containing hydrophilic non-conjugated segments. **B** The molecular structures of hydrophilic segments and the corresponding DCPs.

Supplementary scheme 1 Synthesis procedure of polymers using Pd-catalyzed Suzuki–Miyaura coupling polymerization of fluorene-boronic ester and 3,7-dibromo-5-phenylbenzo[b]phosphindole-5-oxide with either EG-based (EG-Br, TEG-Br, and HEG-Br) or EA-based (EA-Br) segments.

Comment 4. Following on from the above, if I'm not mistaken these are statistical co-polymers, in the sense that the different monomers are not ordered, other than the fluorene unit can never be linked to another fluorene unit, and that there will be a distribution of conjugated and hydrophilic blocks of different lengths. If so, the authors might want to clarify this to the reader and avoid the term block as that would suggest that these are blockpolymers, which these are not.

Response: We are grateful for the reviewer's thoughtful suggestion. Accordingly, we adjusted the "blocks" terminology in the revised manuscript to "segments" to avoid misunderstanding the readers.

Comment 5. I am confused by the statement "the differences between the HOMO/LUMO levels and the optical E_g were small, indicating that hydrophilic non-conjugated building blocks could be molecularly engineered into the main-chain of the target CPs using the present approach without significantly affecting the physicochemical and optical properties". If I'm not mistaken the authors use the E_g value to calculate the LUMO from the measured HOMO level and hence the difference between the LUMO and HOMO levels should be exactly E_g but perhaps I am missing something.

Response: Thank you for drawing our attention to this statement. As mentioned by the reviewer, we did use the E_g value to calculate the LUMO from the measured HOMO level. To avoid the misleading description of our measurement, per the reviewer's suggestion, the original description has been revised on page 7.

"...To sum up, the HOMO/LUMO levels, optical E_g , photoluminescence spectra, and exciton lifetime of all polymers showed no obvious difference, indicating that hydrophilic non-conjugated segments could be molecularly engineered into the main-chain of the target CPs using the present

approach without significantly affecting the physicochemical and optical properties.”

Comment 6. One should be slightly careful with the HOMO and LUMO levels measured by PESA shown in Fig. 2C. Not only are they not truly HOMO and LUMO values but rather more like adiabatic ionisation potentials and electron affinity values (the HOMO/LUMO terminology is problematic when the material has time to relax on the timescale of the photoelectron experiment) but the PESA is done in air while the hydrogen evolution experiments are done in water. When immersing the polymer in water the adiabatic ionisation potentials and electron affinity values will move closer together as water stabilises the charges in the polymer. As a result, a material typically has a smaller thermodynamic driving force for reduction and oxidation than suggested by plotting the PESA derived data versus the half-reaction potentials, as done in Fig. 2C.

Response: Thank you for the reviewer’s thoughtful discussion. Per the reviewer’s suggestion, we also provided the data of cyclic voltammetry (CV) of the polymers in Supplementary Fig. 27 of the revised manuscript. As result, the oxidation potential of these polymers measured by CV exhibited comparable results to photoelectronic spectroscopy.

Polymer	Oxidation Potential (V vs NHE)	$E_{\text{HOMO,CV}}$ (eV)	$E_{\text{HOMO,PESA}}$ (eV)
PFBPO	1.27	-5.67	-5.82
P-EG-5	1.35	-5.74	-5.92
P-TEG-5	1.37	-5.77	-5.91
P-HEG-5	1.33	-5.73	-5.91
P-HEG-10	1.43	-5.83	-5.92
P-HEG-20	1.35	-5.75	-5.90
P-EA-5	1.34	-5.73	-5.93
P-EA-10	1.42	-5.82	-5.91

Supplementary Fig. 27 Cyclic voltammety measurements of **a** PFBPO, **b** P-EG-5, **c** P-TEG-5, **d** P-HEG-5, **e** P-HEG-10, **f** P-HEG-20, **g** P-EA-5, **h** P-EA-10, and **i** ferrocene/ferrocenium couple. Ferrocene/ferrocenium (Fc/Fc⁺) redox potential was measured under the same condition to calibrate the reference electrode. The HOMO levels are determined as follows: $E_{\text{HOMO}} = -(E_{\text{ox, onset}} - E(\text{Fc/Fc}^+) + 4.8)$ eV.

As results, a sentence was added in Page 6 of the revised manuscript.

“Their highest occupied molecular orbital (HOMO) levels ranged from -5.90 to -5.93 eV, as measured using a photoelectronic spectrometer (Supplementary Fig. 26). Furthermore, the oxidation potential of these polymers measured using cyclic voltammety (CV) exhibited comparable results to

photoelectronic spectroscopy (Supplementary Fig. 27).”

Comment 7. What is responsible for the change in contact angle under illumination?

Response: We are grateful for the reviewer’s suggestion. Per the reviewer’s suggestion, we have measured the water contact angle under illumination and found no obvious change in the contact angle before and after illumination. In addition, we are sorry if our time-dependence water contact angle experiments have misunderstood the reviewer. We did measure the water contact angle of polymer films over time to compare the wetting effect of PFBPO and P-HEG-10 films (without light illumination). As shown in Fig. 5e, the decrease in water contact angle of P-HEG-10 is more pronounced than that of PFBPO, indicating that the introduction of hydrophilic segments facilitates the penetration of water into the polymer film. It also demonstrated the advantage of our main-chain engineered DCP as film photocatalysts. To avoid the misleading description of our data, we have rewritten the caption in Fig. 5e of the revised manuscript.

The water contact angle of polymers before and under illumination.

Fig. 5 Photocatalytic hydrogen evolution experiments in film. ...e Time-dependence water contact angles without light illumination of PFBPO and P-HEG-10 films.

Comment 8. I don't think the authors specify the relative proportions of water, TEA and methanol in their reaction mixture for the hydrogen evolution. Is it 1:1:1 on volume basis? Similar what was the ratio between water and TEA in the methanol free experiments (or the TEA concentration)? The reason methanol is typically used is because water and TEA don't mix well at high TEA concentrations and phase separates.

Response: Thanks for your valuable comments. As mentioned by the reviewer, we only described the relative proportions of water, TEA and methanol in the reaction mixture for hydrogen evolution in the Methods paragraph. In a typical photocatalytic experiment, the volume ratio of water, TEA, and methanol is 1:1:1, which means that the reaction mixture consists of 33.3 vol.% H₂O, 33.3 vol.% MeOH, and 33.3 vol.% TEA. While in the methanol-free photocatalytic experiment, the volume ratio of water to TEA was 4:1, which means that the reaction mixture consists of 80 vol.% H₂O and 20 vol.% TEA. Per the reviewer's suggestion, the details of the composition of the photocatalytic reaction mixture have been added to page 12,14,20 in the revised manuscript.

"Mixtures of 33.3 vol.% H₂O, 33.3 vol.% MeOH, and 33.3 vol.% TEA were used, where TEA acts as the sacrificial electron donor and MeOH is added to aid mixing of the TEA with water."

"We further verified the performance of the DCPs for photocatalytic hydrogen evolution in the absence of MeOH (i.e., in an 80 vol.% H₂O/ 20 vol.% TEA solution, Fig. 4e)."

"In a typical measurement, 5 mg of polymer powder and 10 mL of a mixed solution consisting of 33.3 vol.% H₂O, 33.3 vol.% MeOH, and 33.3 vol.% TEA are added to the cuvette and sealed with a septum."

Comment 9. I am slightly not impressed with the fact that only hydrogen evolution data up to 6 hours

is shown. For such a paper, I would have expected to see data for at least the best performing material of 24 hours or more. Other authors report data for 48 hours or more.

Response: We are grateful for the reviewer's suggestion. Long-term photocatalytic hydrogen evolution experiments are crucial to demonstrate the stability and durability of polymer photocatalysts. According to the reviewer's suggestion, we provided a long-term hydrogen evolution data up to 48 h in Supplementary Fig. 48 of the revised manuscript. As results, it demonstrates that our polymer P-10HEG has excellent stability and durability during the photocatalytic reaction.

Supplementary Fig. 48 Long-term photocatalytic cycling up to 48 h was measured to test the durability of P-HEG-10.

As results, a new sentence below was added in the revised manuscript on page 13:

“As shown in Supplementary Fig. 48, the long-term photocatalytic cycling hydrogen evolution evidenced that no any distinct roll-off in the photocatalytic activity for P-HEG-10 during the continuous photocatalytic reaction for 48 h (8 cycles), indicating P-HEG-10 has impressive photo-stability and durability.”

Comment 10. The discussion of the EPR data on page 8 and how it's shown in Supplementary Fig. 27 is potentially confusing. The way I read the data and discussion is that the materials already show an EPR signal in the dark, which is surely not correct. Also, was the EPR done on the dry polymer or the polymer immersed in water and/or the water/methanol/TEA mixture?

Response: Thanks for your valuable comments. All EPRs were performed on dry polymer powders. The EPR spectra shown in Fig. 3c were measured without additional light irradiation (dark). In addition, the EPR spectra shown in Supplementary Fig. 29 displayed the measurements with additional light irradiation, and revealed that the intensity of the signal was further enhanced with additional light irradiation.

In addition, we can observe that many literatures have presented such phenomenon. They reported that their polymers also showed an EPR signal in the dark (without additional light irradiation). [*J. Mater. Chem. A* **3**, 13819 (2015), *J. Mater. Chem. A* **4**, 13166 (2016), *Molecular Catalysis* **453**, 85

(2018), *Polym. Chem.* **10**, 3758 (2019), *Chem. Eur. J.* **25**, 6102 (2019), *Applied Catalysis A: General* **606**, 117833 (2020), *ChemistrySelect* **5**, 14438 (2020), *ACS Appl. Energy Mater.* **5**, 4631 (2022)]

Per the reviewer's suggestion, we considered that the polymers already showed an EPR signal without additional light irradiation because they have responded to light when stored under ambient condition. Therefore, we stored P-HEG-10 in dark for 3 days and then remeasured EPR again without additional light irradiation. Interestingly, although the EPR signal of P-HEG-10 was still observed, the intensity of EPR signal was significantly decreased. This result suggested that the previous EPR signal of the polymers are caused by the light under ambient condition. In addition, we further measured the EPR of the monomers, and found that the EPR signal of the polymer was mainly contributed by the BPO segment instead of the fluorene and HEG segments.

EPR spectra of P-HEG-10 stored under ambient conditions vs. stored in the dark for 3 days.

Following the reviewer's suggestion, we have modified the caption of Fig. 3c and the paragraph on EPR, redrawn the Supplementary Fig. 29 and added the EPR of monomers in Supplementary Fig. 30 of the revised manuscript.

“In the EPR spectra (Fig. 3c and Supplementary Fig. 29), a sharp signal was observed at 3500–3520 G and its intensity increased with additional light irradiation due to more radical generation via additional photoexcitation. The irradiation-induced signal enhancement for the DCPs was greater than that for PFBPO. In addition, we further measured the EPR of the monomers (Supplementary Fig. 30), and found that the EPR signal of the polymer was mainly contributed by the BPO segment instead of the fluorene and HEG segments.”

Fig. 3 Optical and electrochemical properties of polymers. ...c EPR spectra were collected in the solid state under additional light illumination. d...

Supplementary Fig. 29 EPR spectra of **a** PFBPO, **b** P-EG-5, **c** P-TEG-5, **d** P-HEG-5, **e** P-HEG-10, **f** P-HEG-20, **g** P-EA-5, and **h** P-EA-10 were measured without (dash line) and with (solid line) additional light irradiation.

Supplementary Fig. 30 EPR spectra of **a** fluorene, **b** BPO, **c** HEG segments were measured without and with additional light irradiation.

Comment 11. Does Supplementary Fig. 41 show the influence of the Pd co-catalyst on the HER of P-10HEG and PFBPO as the caption suggests or the effect of adding Pt instead?

Response: Thank you for drawing our attention to this issue. We rechecked the figure and corrected the typo in the caption. We study the effect of adding Pt co-catalysts in Supplementary Fig. 49. Therefore, the original caption has been corrected in the revised manuscript.

As results, a revised caption was provided in the revised manuscript:

Supplementary Fig. 49 Influence of the Pt co-catalyst on the HER of P-HEG-10 and PFBPO.

Comment 12. The authors should really share the MD structures used to calculate the data in Fig. 6b and 6C as additional electronic supporting material.

Response: We thank reviewer for pointing this out. Per the reviewer's suggestion, we have provided all the molecular structures in PDB file format required to calculate the data in Fig. 6b and c as well as the table of contents as additional electronic supporting material.

Table of Content

Polymers	PDB file for Fig. 6b	PDB file for Fig. 6c
PFBPO	PFBPO_Fig6b	PFBPO_Fig6c
The hydrophilic segment is at the "end" position.		
P-EG-5	P-EG-5_E_Fig6b	P-EG-5_E_Fig6c
P-TEG-5	P-TEG-5_E_Fig6b	P-TEG-5_E_Fig6c
P-HEG-5	P-HEG-5_E_Fig6b	P-HEG-5_E_Fig6c
P-HEG-10	P-HEG-10_E_Fig6b	P-HEG-10_E_Fig6c
P-HEG-20	P-HEG-20_E_Fig6b	P-HEG-20_E_Fig6c
P-EA-5	P-EA-5_E_Fig6b	P-EA-5_E_Fig6c
P-EA-10	P-EA-10_E_Fig6b	P-EA-10_E_Fig6c
The hydrophilic segment is at the "middle" position.		
P-EG-5	P-EG-5_M_Fig6b	P-EG-5_M_Fig6c
P-TEG-5	P-TEG-5_M_Fig6b	P-TEG-5_M_Fig6c
P-HEG-5	P-HEG-5_M_Fig6b	P-HEG-5_M_Fig6c
P-HEG-10	P-HEG-10_M_Fig6b	P-HEG-10_M_Fig6c
P-HEG-20	P-HEG-20_M_Fig6b	P-HEG-20_M_Fig6c
P-EA-5	P-EA-5_M_Fig6b	P-EA-5_M_Fig6c
P-EA-10	P-EA-10_M_Fig6b	P-EA-10_M_Fig6c
The hydrophilic segment is at the "one-third" position.		
P-EG-5	P-EG-5_1-3_Fig6b	P-EG-5_1-3_Fig6c
P-TEG-5	P-TEG-5_1-3_Fig6b	P-TEG-5_1-3_Fig6c
P-HEG-5	P-HEG-5_1-3_Fig6b	P-HEG-5_1-3_Fig6c
P-HEG-10	P-HEG-10_1-3_Fig6b	P-HEG-10_1-3_Fig6c
P-HEG-20	P-HEG-20_1-3_Fig6b	P-HEG-20_1-3_Fig6c
P-EA-5	P-EA-5_1-3_Fig6b	P-EA-5_1-3_Fig6c
P-EA-10	P-EA-10_1-3_Fig6b	P-EA-10_1-3_Fig6c

Responses to Reviewer #3

Comments to the Author

Comments:

In this work, the authors reported a strategy for improving the photocatalytic hydrogen production activity by introducing hydrophilic non-conjugated building blocks in polymer main-chain and developed a series of organic photocatalysts. A moderate hydrogen evolution rate of 16.8 mmol m⁻² h⁻¹ under visible-light irradiation was obtained by P-10HEG with interrupted conjugation. This work is interesting in the field of photocatalytic hydrogen production by polymer-based photocatalysts. Therefore, this manuscript might be suitable for the publication in Nat. Commun. after the following issues are addressed.

We thank the reviewer for your positive comments. The manuscript was carefully revised based on your suggestions and comments, and we hope that the revised manuscript is suitable for publication in this esteemed journal.

Comment 1. P-10HEG with the longest fluorescence lifetimes has the highest photocatalytic hydrogen evolution rate. It seems that the longer fluorescence lifetimes of the photocatalysts, the higher photocatalytic hydrogen evolution activity. Why does the organic photocatalyst of P-5HEG show a

higher hydrogen evolution rate ($5.38 \pm 0.31 \text{ mmol h}^{-1} \text{ g}^{-1}$) than P-20HEG ($3.74 \pm 0.36 \text{ mmol h}^{-1} \text{ g}^{-1}$), while they have the same fluorescence lifetime (1.11 ns)? Is there a definite relationship between the photocatalytic hydrogen evolution activity and fluorescence lifetime for organic photocatalysts? If not, it is not suitable to evaluate the photocatalytic activity by time-resolved photoluminescence spectroscopy.

Response: Thank you for drawing our attention to this issue. Per the reviewer's suggestion, we carefully remeasured time-resolved photoluminescence data and added the errors in Table 1. To avoid the misleading description of our data, per the reviewer's suggestion, we have added the errors in Table 1 and modified the paragraph on time-resolved photoluminescence on page 7 of the revised manuscript.

“Time-resolved photoluminescence spectroscopy (Fig. 3b) shows that there is no significant difference in exciton lifetimes between all polymers, with their excited-state lifetimes ranging from 0.85–1.07 ns. To sum up, the HOMO/LUMO levels, optical E_g , photoluminescence spectra, and exciton lifetime of all polymers showed no obvious difference, indicating that hydrophilic non-conjugated segments could be molecularly engineered into the main-chain of the target CPs using the present approach without significantly affecting the physicochemical and optical properties.”

Fig. 3 Optical and electrochemical properties of polymers. ...b Time-resolved photoluminescence spectra were obtained in suspensions (5 mg photocatalyst in 10 mL mixture consisting of equal volumes of H_2O , MeOH, and TEA). Samples were excited with a 405 nm laser and emission was measured at 455 nm. **c...**

Comment 2. The authors mentioned that the type and content of hydrophilic non-conjugated building blocks in the hydrophobic CPs do not significantly affect the hydrophilicity of the DCP films due to packing of the polymers. How about the dispersion ability of the polymers in pure water? A dynamic light scattering (DLS) test of the polymers in pure water is suggested to be provided.

Response: We thank the reviewers for their suggestions. Per the reviewer's suggestion, we have measured DLS data of the polymers in pure water and added the results in Supplementary Fig. 44 of the revised manuscript. As results, the polymers showed larger hydrodynamic diameters in pure water

than in the mixture solutions H₂O/MeOH/TEA, because there is no organic solvent to help the dissolution of the polymers. Importantly, in both pure water and mixture solutions, all the DCPs showed larger hydrodynamic diameters than PFBPO.

We further measured DLS data of the polymers in toluene and added the results in Supplementary Fig. 43 of the revised manuscript. We observed that all the polymers were slightly dissolved in toluene, resulting in the smaller hydrodynamic diameters compared to the polymers in the pure water and mixture solutions. Interestingly, all of the DCPs showed smaller hydrodynamic diameters than PFBPO in toluene, indicating that the DCPs have higher solubility in non-aqueous solution. We considered it is because the insertion of the non-conjugated hydrophilic segments to the polymer backbone will reduce the packing of the polymers. Combining the DLS data in these three solvents, the phenomenon that the hydrodynamic diameter of DCPs in both aqueous solutions is larger than that of PFBPO can be truly attributed to swelling in aqueous solutions.

Furthermore, as shown in Fig. 5e, although the PFBPO and P-HEG-10 films exhibited similar initial contact angles, the P-HEG-10 film showed a more significant decrease in contact angle over time than the PFBPO film. Even though the molecular packing of polymer in film state is severe than that in solution state, our main-chain engineered DCP still effectively facilitate water diffuse into the polymer film, demonstrating again that the insertion of this hydrophilic segments into the polymer backbone can truly enhance the penetration of the water.

Supplementary Fig. 43 DLS hydrodynamic diameters of **a** PFBPO, **b** P-EG-5, **c** P-TEG-5, **d** P-HEG-5, **e** P-HEG-10, **f** P-HEG-20, **g** P-EA-5, and **h** P-EA-10 were determined in toluene.

Supplementary Fig. 44 DLS hydrodynamic diameters of **a** PFBPO, **b** P-EG-5, **c** P-TEG-5, **d** P-HEG-5, **e** P-HEG-10, **f** P-HEG-20, **g** P-EA-5, and **h** P-EA-10 were determined in pure water.

Polymer	Size in pure toluene (nm)	Size in H ₂ O/MeOH/TEA (nm)	Size in pure water (nm)
PFBPO	276	471	648
P-EG-5	212	654	815
P-TEG-5	191	674	852
P-HEG-5	188	686	886
P-HEG-10	162	1129	1337
P-HEG-20	152	1576	1618
P-EA-5	182	957	1045
P-EA-10	161	1434	1526

Supplementary Fig. 45 DLS hydrodynamic diameters of a PFBPO, b P-EG-5, c P-TEG-5, d P-HEG-5, e P-HEG-10, f P-HEG-20, g P-EA-5, and h P-EA-10 were determined in a mixture solution consisting of equal volumes of H₂O, MeOH, and TEA.

As results, we have modified the paragraph on page 10 of the revised manuscript:

“... We considered that the severe packing of the polymers in thin film leads to the difference of the water contact angles between the DCPs and PFBPO is not obvious. To further understand the difference of the hydrophilicity between the DCPs and PFBPO, the particle size distributions of the polymers in a non-aqueous (toluene), aqueous (pure water), and mixture solution (33.3 vol.% H₂O/33.3

vol.% MeOH/33.3 vol.% TEA) were determined by dynamic light scattering (DLS). As shown in Supplementary Fig. 43, all of the DCPs showed smaller hydrodynamic diameters than PFBPO in toluene, indicating that the DCPs have higher solubility in non-aqueous solution. We considered it is because the insertion of the non-conjugated hydrophilic segments to the polymer backbone will reduce the packing of the polymers. In addition, the polymers showed larger hydrodynamic diameters in pure water (Supplementary Fig. 44) than in the mixture solutions (Supplementary Fig. 45), because there is no organic solvent to help the dissolution of the polymers. Importantly, in both pure water and mixture solutions, all the DCPs showed larger hydrodynamic diameters than PFBPO due to swelling phenomena in aqueous solution^{29,30}.”

Comment 3. The introduction of hydrophilic chain segment leads to a swelling for the polymer thin-film of P-10HEG after contacting with water, which might be disadvantageous in the photocatalytic stability of polymer thin-film for P-10HEG.

Response: Thanks for your valuable comments. About the photocatalytic stability of polymer thin-film of P-HEG-10, as shown in Fig. 5c, the photocatalytic performance of PFBPO and P-HEG-10 films did not decrease obviously after 6 hours of photocatalytic reaction. In addition, per the reviewer’s suggestion, to further demonstrate the photocatalytic stability of the polymer thin-film of P-HEG-10, we further directly soaked the PFBPO and P-HEG-10 thin-films in photocatalytic solution (the volume ratio of water, TEA, and methanol is 1:1:1) under light illustration for over 120 hours to examine their photocatalytic stability. As shown in Supplementary Fig. 52, both the PFBPO and P-HEG-10 thin-films remain good quality even after 120 hours, demonstrating the introduction of the hydrophilic non-conjugated segments did not significantly affect the photocatalytic stability of polymer thin-film for P-10HEG. We considered that it is owing to that we just inserted a small fraction of the hydrophilic non-conjugated segments into the polymer backbone, which may not affect the stability of polymer thin-film of P-10HEG obviously.

Supplementary Fig. 52 The PFBPO and P-HEG-10 thin-films were directly soaked in photocatalytic

solution (the volume ratio of water, TEA, and methanol is 1:1:1) under light illumination for over 120 hours to examine their photocatalytic stability.

As results, a sentence was added on Page 15 of the revised manuscript.

“To further demonstrate the photocatalytic stability of the polymer thin-films, we directly soaked the PFBPO and P-HEG-10 thin-films in photocatalytic solution under light illumination for over 120 hours. As shown in Supplementary Fig. 52, both the PFBPO and P-HEG-10 thin-films remain good quality even after 120 hours, demonstrating the introduction of small fraction the hydrophilic segments into the polymer backbone did not obviously affect the photocatalytic stability of P-HEG-10 thin-film.”

Comment 4. I have noticed that the samples of hydrophilic P-20HEG and P-10EA show a lower HER than the hydrophobic PFBPO. This should be discussed in detail.

Response: Thanks for your valuable comments. Although hydrophilicity is one of the factors affecting the photocatalytic hydrogen evolution activity, it cannot be ignored the other factors, such as the light absorption of the photocatalytic solution. Previous study has proven that proper aggregation is beneficial to enhance photocatalytic HER, which is related to multiple scattering effects and light absorption [*J. Mater. Chem. A* 7, 2490 (2019)]. Therefore, we considered that the multiple scattering effect and light absorption are the main factors leading to the lower HER of P-HEG-20 and P-EA-10 than PFBPO.

In order to address this issue, as shown in Fig. 4b, we increased the concentrations of photocatalytic solution of P-HEG-20 and then compared with that of PFBPO and P-HEG-10 under the same concentration. The results showed that HER of P-HEG-20 was significantly enhanced compared to that of PFBPO in the same concentration of 10 mg/10 mL and 15 mg/10 mL solution. It demonstrated that even though P-HEG-20 had a lower HER in a condition of 5 mg/10 mL solution, it can present a higher HER than PFBPO in higher concentration. In addition, the lower HER enhancement of PFBPO under the higher concentration also demonstrated that the aggregation of hydrophobic PFBPO was increased when the concentration was increased, so more of the inner polymer chains of the PFBPO cannot interact effectively with water.

Per the reviewer’s suggestion, a detail discussion has added in page 12–13 of the revised manuscript.

“... On the other hand, the HER of P-HEG-20 and P-EA-10 show a lower HER than the hydrophobic PFBPO. We considered that it is owing to the multiple scattering effect and lower light absorption of P-20HEG and P-10EA¹⁷. In order to address this issue, as shown in Fig. 4b, we increased the concentrations of photocatalytic solution of P-HEG-20 and then compared with that of PFBPO. Notably, the result showed that the HER enhancement of P-HEG-10 and P-HEG-20 both are more significant than that of PFBPO under the concentration of 10 mg/10 mL and 15 mg/10 mL solution. (Fig. 4b and Supplementary Table 1). This finding suggests that the lower HER enhancement of PFBPO under the higher concentration also demonstrated that the aggregation of hydrophobic PFBPO was increased when the concentration was increased, so more of the inner polymer chains of the PFBPO cannot interact effectively with water. Excitingly, even though P-HEG-20 had a lower HER in a 5 mg/10 mL solution, P-HEG-20 exhibited a higher HER than PFBPO in the higher solution concentration.”

Fig. 4 Photocatalytic hydrogen evolution experiments in solution. ...b Concentration dependence of the HER over PFBPO, P-HEG-10, and P-HEG-20. c...

Responses to Reviewer #4

Comments to the Author

Comments:

Polymer photocatalysts are an interesting alternative to the more commonly studied inorganic materials for water splitting/hydrogen evolution. A major issue with most polymer photocatalysts is that water has a low level of access to the polymer with large aggregates forming. This is arguably the critical issue in polymer photocatalysis to solve as the water (or electron donor)/polymer interface is likely to be the site of charge separation so minimising exciton transport distances is critical. Furthermore, the water can play an important role in stabilising polymer polaron states. The state of the art is to include hydrophilic groups on conjugated polymers. Here the authors report a really interesting and novel approach which is to disrupt the conjugated structure and put hydrophilic regions into the polymer backbone that break the conjugation.

The concept will be interesting to the community and I encourage its exploration. But the current level experimental evidence linking the levels of photocatalytic activity to the level of polymer hydration/swelling is insufficient to prove the claims. Furthermore, the changes in photocatalytic activity are relatively small (largest increase is 50%).

We thank the reviewer for your positive comments and point out the novelty of this work. The manuscript was carefully revised based on your suggestions and comments, and we hope that the revised manuscript is suitable for publication in this esteemed journal. In the revised manuscript, we have further measured SAXS/WAXS to evaluate the central hypothesis of the swelling effect caused by hydrophilic segments in the DCPs. We also systematically measure and attempt to draw conclusions from DLS experiments in non-aqueous solution and aqueous solution. the detailed results were outlined in Comment 1 below. In addition, in the universality test of our method, we applied this method to another well-known polymer, PF8BT. As results, we found that the change in photocatalytic activity can be achieved up to 235% HER enhancement, which indicates that the enhancement in photocatalytic activity can be varied with the choice of different target polymers. More importantly, we proved that

this novel method can be conveniently introduced to the common hydrophobic polymers to further enhance its initial performance easily.

The main-chain engineering approach was applied to another well-known polymer, PF8BT, and the change in its photocatalytic activity can be achieved up to 235% HER enhancement.

Comment 1. No change in contact angle for water is seen on the polymer films upon inclusion of the hydrophilic groups and instead the authors rely on the particle size as determined by DLS to assess polymer swelling. The particle size in suspension is dependent upon many factors and not just the degree of hydration. To evaluate the papers central hypothesis a more robust measure of polymer hydration is needed.

Response: We thank the reviewers for their suggestions. Per the reviewer's suggestion, we measured wide-range scattering profile covering both the SAXS and WAXS regions and added the result in Fig. 3f of the revised manuscript. As results, fitting of the form factor scattering profile indicated that P-HEG-10 (0.65 nm ± 0.0002 in radius and 11.1 nm ± 0.004 in length) showed an obvious increase in the mean size of the rod-shaped bundles with compared to that of PFBPO (0.46 nm ± 0.0002 in radius and 3.4 nm ± 0.004 in length) in response to the effective expansion of intra-chain distance via hydration/swelling mediated by the hydrophilic segments. In addition, the smearing of the diffraction at $q = 10 \text{ nm}^{-1}$ for P-HEG-10 suggested a perturbation in the ordering of local chain association, which could also be attributed to the hydration/swelling that may locally inhibit the chain packing at atomic length scale.

We also measured DLS data in toluene and added it in Supplementary Fig. 43 of the revised

manuscript. As a result, all polymers were slightly dissolved in toluene and showed smaller hydrodynamic diameters than those measured in the H₂O/MeOH/TEA mixture solution. Importantly, all of the DCPs showed smaller hydrodynamic diameters than PFBPO in toluene, indicating that the DCPs have higher solubility in non-aqueous solution. We considered it is because the insertion of the hydrophilic non-conjugated segments to the polymer backbone will reduce the packing of the polymers.

In addition, we further measured DLS data in pure water and added them in Supplementary Fig. 44 of the revised manuscript. We can find that although the hydrodynamic diameter of the polymer is larger without the help of organic solvent to dissolve, the trend is the same as that measured in the mixed H₂O/MeOH/TEA solution. Importantly, all of the DCPs showed larger hydrodynamic diameters than PFBPO in both aqueous solutions. Combining the DLS data in the three solvents, the phenomenon that the hydrodynamic diameter of DCPs in both aqueous solutions is larger than that of PFBPO can be truly attributed to swelling in aqueous solution.

Interestingly, as shown in Fig. 5e, although the PFBPO and P-HEG-10 films exhibited similar initial contact angles, the P-HEG-10 film showed a more significant decrease in contact angle over time than the PFBPO film. Even though the molecular packing of polymer in film state is severe than that in solution state, our main-chain engineered DCP still effectively facilitate water diffuse into the polymer film, demonstrating again that the insertion of this hydrophilic segments into the polymer backbone can truly enhance the penetration of the water.

Fig. 3 Optical and electrochemical properties of polymers. ...f Wide-angle X-ray scattering profiles of PFBPO and P-HEG-10 were measured in a mixture solution consisting of equal volumes of H₂O, MeOH, and TEA.

As results, a new sentence below was added in the revised manuscript on page 11:

“Interesting, the level of hydration/swelling in P-HEG-10 could be evidenced by an obvious increase in the mean size of the rod-shaped bundles as resolved by the wide-range scattering profile covering both the SAXS and WAXS regions with compared to that of PFBPO (Fig. 3f). It was clearly shown that the form factor ($q = 0.3\text{--}4 \text{ nm}^{-1}$) contributed from the rod-like bundles of P-HEG-10 located at the q range lower than that of PFBPO ($q = 0.4\text{--}4 \text{ nm}^{-1}$), indicating that the average size of the bundles

formed in P-HEG-10 was larger than PFBPO in response to the effective expansion of intra-chain distance via hydration/swelling mediated by the hydrophilic segments. Comparison between the fitted curves of rod form factor scatterings revealed that the geometric sizes of P-HEG-10 bundles ($0.65 \text{ nm} \pm 0.0002$ in radius and $11.1 \text{ nm} \pm 0.004$ in length) were indeed larger than that of PFBPO bundles ($0.46 \text{ nm} \pm 0.0002$ in radius and $3.4 \text{ nm} \pm 0.004$ in length). Particularly, the size expansion of P-HEG-10 bundles in the radial direction seems to be beneficial to increase the length of the bundles. It could be deduced that absorbed water molecules inside the bundles were able to act as structure-directing agent to induce more chain segments to be involved in packing along the axial direction of the bundles. This shall enhance the specific surface area from the aligned polymer bundles for higher photocatalytic activity relative to the entangled state. A further investigation into the smearing of the diffraction at $q = 10 \text{ nm}^{-1}$ for P-HEG-10 suggested a perturbation in the ordering of local chain association, which could also be attributed to the hydration/swelling that may locally inhibit the chain packing at atomic length scale.”

Supplementary Fig. 43 DLS hydrodynamic diameters of **a** PFBPO, **b** P-EG-5, **c** P-TEG-5, **d** P-HEG-5, **e** P-HEG-10, **f** P-HEG-20, **g** P-EA-5, and **h** P-EA-10 were determined in toluene.

Supplementary Fig. 44 DLS hydrodynamic diameters of **a** PFBPO, **b** P-EG-5, **c** P-TEG-5, **d** P-HEG-5, **e** P-HEG-10, **f** P-HEG-20, **g** P-EA-5, and **h** P-EA-10 were determined in pure water.

Polymer	Size in pure toluene (nm)	Size in H ₂ O/MeOH/TEA (nm)	Size in pure water (nm)
PFBPO	276	471	648
P-EG-5	212	654	815
P-TEG-5	191	674	852
P-HEG-5	188	686	886
P-HEG-10	162	1129	1337
P-HEG-20	152	1576	1618
P-EA-5	182	957	1045
P-EA-10	161	1434	1526

Supplementary Fig. 45 DLS hydrodynamic diameters of a PFBPO, b P-EG-5, c P-TEG-5, d P-HEG-5, e P-HEG-10, f P-HEG-20, g P-EA-5, and h P-EA-10 were determined in a mixture solution consisting of equal volumes of H₂O, MeOH, and TEA.

As results, we have modified the paragraph on page 10 of the revised manuscript:

“... We considered that the severe packing of the polymers in thin film leads to the difference of the water contact angles between the DCPs and PFBPO is not obvious. To further understand the difference of the hydrophilicity between the DCPs and PFBPO, the particle size distributions of the polymers in a non-aqueous (toluene), aqueous (pure water), and mixture solution (33.3 vol.% H₂O/33.3

vol.% MeOH/33.3 vol.% TEA) were determined by dynamic light scattering (DLS). As shown in Supplementary Fig. 43, all of the DCPs showed smaller hydrodynamic diameters than PFBPO in toluene, indicating that the DCPs have higher solubility in non-aqueous solution. We considered it is because the insertion of the non-conjugated hydrophilic segments to the polymer backbone will reduce the packing of the polymers. In addition, the polymers showed larger hydrodynamic diameters in pure water (Supplementary Fig. 44) than in the mixture solutions (Supplementary Fig. 45), because there is no organic solvent to help the dissolution of the polymers. Importantly, in both pure water and mixture solutions, all the DCPs showed larger hydrodynamic diameters than PFBPO due to swelling phenomena in aqueous solution^{29,30}.”

Comment 2. There is no correlation between particle sizes and photocatalytic activity. For example, P-20HEG has the largest particle size but a photocatalytic activity that is less than the control material. This is perhaps not surprising as noted above, the DLS is not a good measure of hydration.

Response: Thanks for your valuable comments. Although hydrophilicity is one of the factors affecting the photocatalytic hydrogen evolution activity, it cannot be ignored the other factors, such as the light absorption of the photocatalytic solution. Previous study has proven that proper aggregation is beneficial to enhance photocatalytic HER, which is related to multiple scattering effects and light absorption [*J. Mater. Chem. A* 7, 2490 (2019)]. Therefore, we considered that the multiple scattering effect and light absorption are the main factors leading to the lower HER of P-HEG-20 and P-EA-10 than PFBPO.

Therefore, we considered that the multiple scattering effect and light absorption are the main factors leading to the lower HER of P-HEG-20 and P-EA-10 than PFBPO. In order to address this issue, as shown in Fig. 4b, we increased the concentrations of photocatalytic solution of P-HEG-20 and then compared with PFBPO and P-HEG-10. The results showed that HER of P-HEG-20 was significantly enhanced compared to that of PFBPO in the concentration of 10 mg/10 mL and 15 mg/10 mL solution. It demonstrated that even though P-HEG-20 had a lower HER in a condition of 5 mg/10 mL solution, it can present a higher HER than PFBPO in higher concentration. In addition, the lower HER enhancement of PFBPO under the higher concentration also demonstrated that the aggregation of hydrophobic PFBPO was increased when the concentration was increased, so more of the inner polymer chains of the PFBPO cannot interact effectively with water.

Per the reviewer’s suggestion, a detail discussion has added in page 12–13 of the revised manuscript.

“...On the other hand, the HER of P-HEG-20 and P-EA-10 show a lower HER than the hydrophobic PFBPO. We considered that it is owing to the multiple scattering effect and lower light absorption of P-20HEG and P-10EA¹⁷. In order to address this issue, as shown in Fig. 4b, we increased the concentrations of photocatalytic solution of P-HEG-20 and then compared with that of PFBPO. Notably, the result showed that the HER enhancement of P-HEG-10 and P-HEG-20 both are more significant than that of PFBPO under the concentration of 10 mg/10 mL and 15 mg/10 mL solution. (Fig. 4b and Supplementary Table 1). This finding suggests that the lower HER enhancement of PFBPO under the higher concentration also demonstrated that the aggregation of hydrophobic PFBPO was increased when the concentration was increased, so more of the inner polymer chains of the PFBPO cannot interact effectively with water. Excitingly, even though P-HEG-20 had a lower HER in a 5 mg/10

mL solution, P-HEG-20 exhibited a higher HER than PFBPO in the higher solution concentration.”

Fig. 4 Photocatalytic hydrogen evolution experiments in solution. ...b Concentration dependence of the HER over PFBPO, P-HEG-10, and P-HEG-20. c...

Comment 3. Data is presented for only a small selection of the materials for several of the important experiments. For the EPR and PL quenching data for P-5EG, P10-HEG and P-5EA is provided but it is not clear why these samples are chosen as they do not allow for assessment of percentage of hydrophilic group on behaviour or comparison between the hydrophilic groups. For the CV only P-10HEG is shown.

Response: Thanks for your valuable comments. Per the reviewer's suggestion, we added the EPR, PL quenching, and CV data for all polymers and modified all paragraphs about them in the revised manuscript.

Fig. 3 Optical and electrochemical properties of polymers. ...c EPR spectra were collected in the solid state under additional light illumination. d...

As results, we have modified the paragraph on EPR on page 8 of the revised manuscript:

“In the EPR spectra (Fig. 3c and Supplementary Fig. 29), a sharp signal was observed at 3500–

3520 G and its intensity increased with additional light irradiation due to more radical generation via additional photoexcitation. The irradiation-induced signal enhancement for the DCPs was greater than that for PFBPO. In addition, we further measured the EPR of the monomers (Supplementary Fig. 30), and found that the EPR signal of the polymer was mainly contributed by the BPO segment instead of the fluorene and HEG segments.”

Supplementary Fig. 28 Photoluminescence spectra of **a** PFBPO, **b** P-EG-5, **c** P-TEG-5, **d** P-HEG-5, **e** P-HEG-10, **f** P-HEG-20, **g** P-EA-5, and **h** P-EA-10 were obtained in suspensions (5 mg photocatalyst in 10 mL mixture consisting of equal volumes of H₂O, MeOH, and TEA) before (solid line) and after (dash line) adding 3 wt.% H₂PtCl₆ co-catalyst, respectively, in which the pH keeps consistent at 12.4.

As results, we have modified the paragraph on PL on page 6 of the revised manuscript:

“As shown in Fig. 3a, the emission maxima of all polymers fall at almost the same wavelength. Furthermore, all polymers were effectively quenched by the addition of Pt co-catalyst (Supplementary Fig. 28). The photoluminescence quenching degree of DCPs was higher than that of PFBPO, indicating improved charge transfer from DCPs to the Pt co-catalyst.”

Polymer	Oxidation Potential (V vs NHE)	$E_{\text{HOMO,CV}}$ (eV)	$E_{\text{HOMO,PESA}}$ (eV)
PFBPO	1.27	-5.67	-5.82
P-EG-5	1.35	-5.74	-5.92
P-TEG-5	1.37	-5.77	-5.91
P-HEG-5	1.33	-5.73	-5.91
P-HEG-10	1.43	-5.83	-5.92
P-HEG-20	1.35	-5.75	-5.90
P-EA-5	1.34	-5.73	-5.93
P-EA-10	1.42	-5.82	-5.91

Supplementary Fig. 27 Cyclic voltammetry measurements of **a** PFBPO, **b** P-EG-5, **c** P-TEG-5, **d** P-HEG-5, **e** P-HEG-10, **f** P-HEG-20, **g** P-EA-5, **h** P-EA-10, and **i** ferrocene/ferrocenium couple. Ferrocene/ferrocenium (Fc/Fc⁺) redox potential was measured under the same condition to calibrate the reference electrode. The HOMO levels are determined as follows: $E_{\text{HOMO}} = -(E_{\text{ox, onset}} - E(\text{Fc/Fc}^+) + 4.8)$ eV.

As results, we have modified the paragraph on CV on page 6 of the revised manuscript:

“Their highest occupied molecular orbital (HOMO) levels ranged from -5.90 to -5.93 eV, as measured using a photoelectronic spectrometer (Supplementary Fig. 26). Furthermore, the oxidation potential of these polymers measured using cyclic voltammetry (CV) exhibited comparable results to

photoelectronic spectroscopy (Supplementary Fig. 27).”

Comment 4. The CV of P0-HEG has a redox couple at ~ -0.7 V Ag/AgCl. This is -0.5 Vs NHE. Assuming the redox couple is pH independent this means the driving force for H₂ evolution is ~ 250 meV at pH 12.7, significantly less than Fig 2 indicates. This is perhaps ok as the XPS and optical gap is reporting on the vertical transition and the CV on the potential of polymer reduction. But it would be useful to show the CV's for the other polymers and also to check the pH dependence of the redox couple.

Response: Thanks for your valuable comments. Many reports demonstrated that the redox couple is pH dependent, and observed the Nernstian behavior from the electrochemical experiments [*J. Chem. Educ.* **74**, 1195 (1997); *Sci. Rep.* **9**, 4537 (2019); *Analyst* **144**, 1386 (2019); *J. Electroanal. Chem.* **845**, 1 (2019); *Polymers* **12**, 2328 (2020); *RSC Adv.* **10**, 28454 (2020); *J. Electroanal. Chem.* **895**, 115530 (2021)]. Therefore, the discussion of electrochemical potential should be based on the same measurement condition or calibrated using internal standard and equation.

Per the reviewer's suggestion, we have carefully rechecked and remeasured cyclic voltammetry (CV) spectroscopy, and added the results in Supplementary Fig. 27. For calibration, the redox potential of ferrocene/ferrocenium (Fc/Fc⁺) was measured under the same conditions, and it is located at 0.08 V to the Ag/AgNO₃ electrode. After calculating using an absolute energy level of the redox potential of Fc/Fc⁺ (4.80 eV to vacuum) and optical band gap obtained from Tauc plot, we can determine the highest occupied molecular orbital (HOMO) and lowest unoccupied molecular orbital (LUMO) level of all polymer were in the range of 5.67–5.83 eV to vacuum and 2.98–3.07 eV to vacuum, indicating that the driving force for H₂ production ($E_{\text{abs.}} = \sim 4.44$ eV to vacuum [*J. Electroanal. Chem. Interfacial Electrochem.* **209**, 417 (1986); *Chem. Sci.* **5**, 1216 (2014)]) is at least larger than 1.37 eV.

Polymer	Oxidation Potential (V vs NHE)	$E_{\text{HOMO,CV}}$ (eV)	$E_{\text{HOMO,PESA}}$ (eV)
PFBPO	1.27	-5.67	-5.82
P-EG-5	1.35	-5.74	-5.92
P-TEG-5	1.37	-5.77	-5.91
P-HEG-5	1.33	-5.73	-5.91
P-HEG-10	1.43	-5.83	-5.92
P-HEG-20	1.35	-5.75	-5.90
P-EA-5	1.34	-5.73	-5.93
P-EA-10	1.42	-5.82	-5.91

Supplementary Fig. 27 Cyclic voltammetry measurements of **a** PFBPO, **b** P-EG-5, **c** P-TEG-5, **d** P-HEG-5, **e** P-HEG-10, **f** P-HEG-20, **g** P-EA-5, **h** P-EA-10, and **i** ferrocene/ferrocenium couple. Ferrocene/ferrocenium (Fc/Fc^+) redox potential was measured under the same condition to calibrate the reference electrode. The HOMO levels are determined as follows: $E_{\text{HOMO}} = -(E_{\text{ox, onset}} - E(\text{Fc}/\text{Fc}^+) + 4.8)$ eV.

As results, we have modified the paragraph on CV on page 6 of the revised manuscript:

“Their highest occupied molecular orbital (HOMO) levels ranged from -5.90 to -5.93 eV, as measured using a photoelectronic spectrometer (Supplementary Fig. 26). Furthermore, the oxidation potential of these polymers measured using cyclic voltammetry (CV) exhibited comparable results to

photoelectronic spectroscopy (Supplementary Fig. 27).”

Comment 5. UV/Vis spectra show that several of the polymers have a broad feature at long-wavelengths which is not described. Fig 4d shows it is photo-catalytically active transition so it is important to discuss.

Response: Thank you for drawing our attention to this issue. Per the reviewer's suggestion, we have carefully rechecked and remeasured UV/Vis absorption spectra in both powder and solution state. As results, all polymer did not exhibit a broad absorption band at long-wavelength. Therefore, we revised the Fig. 2a in the revised manuscript.

Fig. 2 Optical properties of polymers. a Solid-state UV-vis diffuse reflectance spectra, **b...**

The absorption spectra of the polymers in solution state.

Comment 6. Fig 3 b shows the lifetimes of the PL are essentially the same but table 1 lists quite large differences in the lifetime (but no errors). All values look like they are very close to the instrument response function. I caution against over interpretation of these.

Response: Thank you for drawing our attention to this issue. Per the reviewer's suggestion, we carefully remeasured time-resolved photoluminescence data and added the errors in Table 1. To avoid the

misleading description of our data, per the reviewer's suggestion, we have added the errors in Table 1 and modified the paragraph on time-resolved photoluminescence on page 7 of the revised manuscript.

“Time-resolved photoluminescence spectroscopy (Fig. 3b) shows that there is no significant difference in exciton lifetimes between all polymers, with their excited-state lifetimes ranging from 0.85–1.07 ns. To sum up, the HOMO/LUMO levels, optical E_g , photoluminescence spectra, and exciton lifetime of all polymers showed no obvious difference, indicating that hydrophilic non-conjugated segments could be molecularly engineered into the main-chain of the target CPs using the present approach without significantly affecting the physicochemical and optical properties.”

Fig. 3 Optical and electrochemical properties of polymers. ...b Time-resolved photoluminescence spectra were obtained in suspensions (5 mg photocatalyst in 10 mL mixture consisting of equal volumes of H₂O, MeOH, and TEA). Samples were excited with a 405 nm laser and emission was measured at 455 nm. c...

Comment 7. Page 8, line 163. EIS shows a lower resistance which indicates enhanced charge transport. Why would increasing the water/polymer interface area improve charge transport? I expect the primary role of the water is to increase charge separation yields. I would have expected disruption of the conjugation would decrease charge transport.

Response: Thank you for your valuable comment. Per the reviewer's suggestion, we added the equivalent circuit in Fig. 3d and Supplementary Fig. 31 and have rewritten the caption in Fig. 3d of the revised manuscript. In the equivalent circuit, R_s represents the circuit series-resistance, R_{ct} is the charge transfer resistance across the interface, and C_{dl} is the capacitance phase element of the semiconductor-electrolyte interface. In addition, the circuit series-resistance (R_s) and charge transfer resistance (R_{ct}) measurements of prepared photocatalysts were added in Supplementary Fig. 31. There is no significant change in the R_s value, including the electrolyte resistance, polymer resistance and contact resistance between the photocatalysts and FTO substrate. On the other hand, R_{ct} is decreased with an increase of hydrophilic non-conjugated segments, considering the enhancement of the interface charge transfer.

Fig. 3 ...d Electrochemical impedance spectra of polymers were carried out in dark with an AC potential frequency ranging from 0.1 Hz to 100 kHz. In the equivalent circuit, R_s represents the circuit series-resistance, R_{ct} is the charge transfer resistance across the interface, and C_{dl} is the capacitance phase element of the semiconductor-electrolyte interface. e...

Polymer	R_s (Ω)	R_{ct} (Ω)
PFBPO	1552	5128
P-EG-5	1574	4668
P-TEG-5	1618	4436
P-HEG-5	1635	4252
P-HEG-10	1654	4104
P-HEG-20	1658	3942
P-EA-5	1668	4056
P-EA-10	1651	3984

Supplementary Fig. 31 Electrochemical impedance spectra of polymers were carried out in dark with an AC potential frequency ranging from 0.1 Hz to 100 kHz. In the equivalent circuit (inset), R_s represents the circuit series-resistance, R_{ct} is the charge transfer resistance across the interface, and C_{dl} is the capacitance phase element of the semiconductor-electrolyte interface. Simulated R_s and R_{ct} values of polymers for electrochemical impedance test were list in table.

Therefore, we added the paragraph on EIS on page 8 of the revised manuscript.

“The EIS data recorded in the dark showed a semicircular curve (Fig. 3d and Supplementary Fig. 31), where the charge transfer resistance values of DCPs is lower than that of PFBPO due to the incorporation of hydrophilic non-conjugated segments, suggesting that the DCPs have a more suitable interface for charge transfer in photocatalytic hydrogen production reaction.”

Responses to Reviewer #5

Comments to the Author

Comments:

In this paper, the authors report a new type of conjugated polymer for use in photocatalysis. The authors provide characterization of the materials, as well as their photocatalytic activity and ultimately, conclude that the amount of hydrogen bonding of the system with water dictates the efficacy towards water splitting. While the work is interesting, it is not immediately apparent to me that the work provides new insight in terms of using conjugated polymers for water splitting for reasons primarily outlined in point 1 below. There is also a little bit of data missing which makes it difficult to judge some of the claims made in the paper.

Thanks for the reviewer's comments. The manuscript was carefully revised based on the suggestions and comments, and we hope that the revised manuscript is suitable for publication in this esteemed journal.

Comment 1. The authors indicate that this is the first time that a system with broken conjugation was shown to be effective in water splitting. However, I disagree with this concept because Figure 2 shows that all the UV-vis spectra are similar. This similarity indicates that the effective conjugation length of all systems is similar, which in turn means that the conjugation was not broken or that the parent PFBPO has broken conjugation in the first place. In that sense, this claim is misleading.

Response: Thanks for the reviewer's comments. In this study, we have shown the molecular weight of all polymers by GPC measurement, and the results of the M_n of the polymers are about 10.4–15.2 kDa, which clearly indicated that both PFBPO and DCPs were successfully polymerized. In addition, the ^1H NMR spectra of DCP showed the characteristic signals of the EG- and EA-based segments at 3.5–4.5 ppm, suggesting that non-conjugated hydrophilic segments were truly presented in the DCP backbones. Moreover, if the hydrophilic segments are not in the polymer backbone, then it cannot show the many different behaviors such as their photocatalytic performance, the DLS in aqueous or non-aqueous solution, the SAXS/WAXS and so on that we showed in the manuscript.

Notably, we also provided the absorption spectra of the monomers (as the figure shown below), and it is clearly observed that both PFBPO and P-HEG-10 showed more red-shifted absorption band compared to the monomers. If the parent PFBPO has broken conjugation length, the optical properties of the polymer should be very similar to its monomers, if not, then it means the conjugation length of PFBPO is effective. The aforementioned results strongly demonstrated that our parent PFBPO does not have broken conjugation in the first place and has reached to its effective conjugation length. Furthermore, our DCP series has similar absorption band with PFBPO, also demonstrating DCPs can reach the similar effective conjugation length to the parent PFBPO, when inserting the small fraction of non-conjugated hydrophilic segments to the PFBPO backbone. Again, if this small fraction of non-conjugated hydrophilic segments gives the conjugation length become non-effectively, then the optical properties of DCP should be not that similar to that of PFBPO. We can see the figure as below; they have very similar absorption band actually. All these results proved that our samples for following studies are well-conjugated polymers.

The absorption spectra of monomers (F8-B, BPO-Br and HEG-Br) and polymers (PFBPO and P-HEG-10) in solution state.

When we further applied the main-chain engineering approach to other published polymers, PF8BT and PCPDTBSO, we also observed similar absorption spectra between pristine polymer and main-chain-engineered DCP. All three polymer systems proved that hydrophilic non-conjugated segments could be molecularly engineered into the main-chain of the target CPs using the present approach without significantly affecting the physicochemical and optical properties.

In the field of using conjugated polymers for organic electronics, previous reports also demonstrated that introducing a small fraction of non-conjugated units to the polymer backbone does not noticeably degrade their semiconducting or optoelectronic properties (Nature 539, 411 (2016); Adv. Funct. Mater. 28, 1804222 (2018); Macromolecules 51, 5944 (2018); ACS Appl. Polym. Mater. 1, 315 (2019); Chem. Mater. 32, 5700 (2020); Macromolecules 54, 7388 (2021); Energy Environ. Sci. 14, 4067 (2021); Adv. Funct. Mater. 32, 2106564 (2022)). According to these previous reports and our experimental results in this study, although the introduction of hydrophilic non-conjugated segments into the main-chain of conjugated polymer interrupted the conjugation, but the use of small fraction of the hydrophilic segments did not obviously affect the semiconducting properties the prepared polymers.

In addition, our approach can address a major issue with most polymer photocatalysts is that water has a low level of access to the polymer with large aggregates forming. The hydrophilic non-conjugated segments successfully produced excellent water/polymer interface and effectively brought water into the inner polymer chain to enhance the photocatalytic performance without obviously changing the semiconducting properties of the polymers. Therefore, the key point of this work is that we use a new main-chain engineering method of polymer photocatalysts to enhance the water/polymer interface, to further minimize the exciton transport distances. We also apply this method to various polymers with different optical bandgap (as the figures shown below), and obtained the enhanced photocatalytic hydrogen evolution results in all system, suggesting that our method is new and universal.

The absorption spectra of pristine polymers and main-chain-engineered DCPs in solution state for three polymer systems.

Comment 2. For the EIS data shown on Page 9, the authors should indicate what equivalent circuit was used to calculate the resistance.

Response: We thank the reviewers for their suggestions. Per the reviewer's suggestion, we have added the equivalent circuit in Fig. 3d and Supplementary Fig. 31, and have rewritten the caption in Fig. 3d of the revised manuscript. In the equivalent circuit, R_s represents the circuit series-resistance, R_{ct} is the charge transfer resistance across the interface, and C_{dl} is the capacitance phase element of the semiconductor-electrolyte interface.

Fig. 3 ...d Electrochemical impedance spectra of polymers were carried out in dark with an AC potential frequency ranging from 0.1 Hz to 100 kHz. In the equivalent circuit, R_s represents the circuit series-resistance, R_{ct} is the charge transfer resistance across the interface, and C_{dl} is the capacitance phase element of the semiconductor-electrolyte interface. e...

Comment 3. Page 8: “The EIS data recorded in the dark showed a semicircular curve (Fig. 3d), where the DCPs had resistance values lower than that of PFBPO due to their enhanced charge transport” The data, as shown, is inconclusive to support that the DCPs had enhanced charge transport. Related to point 2 above, more discussion needs to be provided as to how the authors determined if the arc shown in the Nyquist plot is DCP resistance, not electrolyte or interface resistance.

Response: Thank you for your valuable comment. Per the reviewer's suggestion, the circuit series-

resistance (R_s) and charge transfer resistance (R_{ct}) measurements of prepared photocatalysts were added in Supplementary Fig. 31. There is no significant change in the R_s value, including the electrolyte resistance, polymer resistance and contact resistance between the photocatalysts and FTO substrate. On the other hand, R_{ct} is decreased with an increase of hydrophilic non-conjugated segments, presumably enhancing the interface charge transfer.

Polymer	R_s (Ω)	R_{ct} (Ω)
PFBPO	1552	5128
P-EG-5	1574	4668
P-TEG-5	1618	4436
P-HEG-5	1635	4252
P-HEG-10	1654	4104
P-HEG-20	1658	3942
P-EA-5	1668	4056
P-EA-10	1651	3984

Supplementary Fig. 31 Electrochemical impedance spectra of polymers were carried out in dark with an AC potential frequency ranging from 0.1 Hz to 100 kHz. In the equivalent circuit (inset), R_s represents the circuit series-resistance, R_{ct} is the charge transfer resistance across the interface, and C_{dl} is the capacitance phase element of the semiconductor-electrolyte interface. Simulated R_s and R_{ct} values of polymers for electrochemical impedance test were list in table.

Therefore, we added the paragraph on EIS on page 8 of the revised manuscript:

“The EIS data recorded in the dark showed a semicircular curve (Fig. 3d and Supplementary Fig. 31), where the charge transfer resistance values of DCPs is lower than that of PFBPO due to the incorporation of hydrophilic non-conjugated segments, suggesting that the DCPs have a more suitable interface for charge transfer in photocatalytic hydrogen production reaction.”

Comment 4. For Figure S38, please indicate in the figure caption what solvent was used for the DLS measurement.

Response: Thank you for your valuable comments. Per the reviewer’s suggestion, we added the measurement condition of DLS in Supplementary Fig. 45 and revised the caption of Supplementary Fig. 45.

As results, a new caption was provided in the revised manuscript:

Polymer	Size in pure toluene (nm)	Size in H ₂ O/MeOH/TEA (nm)	Size in pure water (nm)
PFBPO	276	471	648
P-EG-5	212	654	815
P-TEG-5	191	674	852
P-HEG-5	188	686	886
P-HEG-10	162	1129	1337
P-HEG-20	152	1576	1618
P-EA-5	182	957	1045
P-EA-10	161	1434	1526

Supplementary Fig. 45 DLS hydrodynamic diameters of **a** PFBPO, **b** P-EG-5, **c** P-TEG-5, **d** P-HEG-5, **e** P-HEG-10, **f** P-HEG-20, **g** P-EA-5, and **h** P-EA-10 were determined in a mixture solution consisting of equal volumes of H₂O, MeOH, and TEA.

Comment 5. Pages 9-10: The DLS results are discussed here and it is postulated that the DCPs swelled in an aqueous solution. Can the authors also provide the DLS data in a non-aqueous solution too to support that the bigger size is indeed due to swelling instead of being an intrinsic property of the particles?

Response: We thank the reviewers for their suggestions. Per the reviewer's suggestion, we measured

DLS data in toluene and added it in Supplementary Fig. 43 of the revised manuscript. As a result, all polymers were slightly dissolved in toluene and showed smaller hydrodynamic diameters than those measured in the H₂O/MeOH/TEA mixture solution. Importantly, all of the DCPs showed smaller hydrodynamic diameters than PFBPO in toluene, indicating that the DCPs have higher solubility in non-aqueous solution. We considered it is because the insertion of the hydrophilic non-conjugated segments to the polymer backbone will reduce the packing of the polymers.

In addition, we further measured DLS data in pure water and added them in Supplementary Fig. 44 of the revised manuscript. We can find that although the hydrodynamic diameter of the polymer is larger without the help of organic solvent to dissolve, the trend is the same as that measured in the mixed H₂O/MeOH/TEA solution. Importantly, all of the DCPs showed larger hydrodynamic diameters than PFBPO in both aqueous solutions. Combining the DLS data in the three solvents, the phenomenon that the hydrodynamic diameter of DCPs in both aqueous solutions is larger than that of PFBPO can be truly attributed to swelling in aqueous solution.

Supplementary Fig. 43 DLS hydrodynamic diameters of **a** PFBPO, **b** P-EG-5, **c** P-TEG-5, **d** P-HEG-5, **e** P-HEG-10, **f** P-HEG-20, **g** P-EA-5, and **h** P-EA-10 were determined in toluene.

Supplementary Fig. 44 DLS hydrodynamic diameters of **a** PFBPO, **b** P-EG-5, **c** P-TEG-5, **d** P-HEG-5, **e** P-HEG-10, **f** P-HEG-20, **g** P-EA-5, and **h** P-EA-10 were determined in pure water.

Polymer	Size in pure toluene (nm)	Size in H ₂ O/MeOH/TEA (nm)	Size in pure water (nm)
PFBPO	276	471	648
P-EG-5	212	654	815
P-TEG-5	191	674	852
P-HEG-5	188	686	886
P-HEG-10	162	1129	1337
P-HEG-20	152	1576	1618
P-EA-5	182	957	1045
P-EA-10	161	1434	1526

Supplementary Fig. 45 DLS hydrodynamic diameters of a PFBPO, b P-EG-5, c P-TEG-5, d P-HEG-5, e P-HEG-10, f P-HEG-20, g P-EA-5, and h P-EA-10 were determined in a mixture solution consisting of equal volumes of H₂O, MeOH, and TEA.

As results, we have modified the paragraph on page 10 of the revised manuscript:

“... We considered that the severe packing of the polymers in thin film leads to the difference of the water contact angles between the DCPs and PFBPO is not obvious. To further understand the difference of the hydrophilicity between the DCPs and PFBPO, the particle size distributions of the polymers in a non-aqueous (toluene), aqueous (pure water), and mixture solution (33.3 vol.% H₂O/33.3

vol.% MeOH/33.3 vol.% TEA) were determined by dynamic light scattering (DLS). As shown in Supplementary Fig. 43, all of the DCPs showed smaller hydrodynamic diameters than PFBPO in toluene, indicating that the DCPs have higher solubility in non-aqueous solution. We considered it is because the insertion of the non-conjugated hydrophilic segments to the polymer backbone will reduce the packing of the polymers. In addition, the polymers showed larger hydrodynamic diameters in pure water (Supplementary Fig. 44) than in the mixture solutions (Supplementary Fig. 45), because there is no organic solvent to help the dissolution of the polymers. Importantly, in both pure water and mixture solutions, all the DCPs showed larger hydrodynamic diameters than PFBPO due to swelling phenomena in aqueous solution^{29,30}.”

In addition, we also measured wide-range scattering profile covering both the SAXS and WAXS regions and added the result in Fig. 3f of the revised manuscript. As results, fitting of the form factor scattering profile indicated that P-HEG-10 (0.65 nm ± 0.0002 in radius and 11.1 nm ± 0.004 in length) showed an obvious increase in the mean size of the rod-shaped bundles with compared to that of PFBPO (0.46 nm ± 0.0002 in radius and 3.4 nm ± 0.004 in length) in response to the effective expansion of intra-chain distance via hydration/swelling mediated by the hydrophilic segments. In addition, the smearing of the diffraction at $q = 10 \text{ nm}^{-1}$ for P-HEG-10 suggested a perturbation in the ordering of local chain association, which could also be attributed to the hydration/swelling that may locally inhibit the chain packing at atomic length scale.

Fig. 3 Optical and electrochemical properties of polymers. ...f Wide-angle X-ray scattering profiles of PFBPO and P-HEG-10 were measured in a mixture solution consisting of equal volumes of H₂O, MeOH, and TEA.

As results, a new sentence below was added in the revised manuscript on page 11:

“Interesting, the level of hydration/swelling in P-HEG-10 could be evidenced by an obvious increase in the mean size of the rod-shaped bundles as resolved by the wide-range scattering profile covering both the SAXS and WAXS regions with compared to that of PFBPO (Fig. 3f). It was clearly shown that the form factor ($q = 0.3\text{--}4 \text{ nm}^{-1}$) contributed from the rod-like bundles of P-HEG-10 located at the q range lower than that of PFBPO ($q = 0.4\text{--}4 \text{ nm}^{-1}$), indicating that the average size of the bundles

formed in P-HEG-10 was larger than PFBPO in response to the effective expansion of intra-chain distance via hydration/swelling mediated by the hydrophilic segments. Comparison between the fitted curves of rod form factor scatterings revealed that the geometric sizes of P-HEG-10 bundles ($0.65 \text{ nm} \pm 0.0002$ in radius and $11.1 \text{ nm} \pm 0.004$ in length) were indeed larger than that of PFBPO bundles ($0.46 \text{ nm} \pm 0.0002$ in radius and $3.4 \text{ nm} \pm 0.004$ in length). Particularly, the size expansion of P-HEG-10 bundles in the radial direction seems to be beneficial to increase the length of the bundles. It could be deduced that absorbed water molecules inside the bundles were able to act as structure-directing agent to induce more chain segments to be involved in packing along the axial direction of the bundles. This shall enhance the specific surface area from the aligned polymer bundles for higher photocatalytic activity relative to the entangled state. A further investigation into the smearing of the diffraction at $q = 10 \text{ nm}^{-1}$ for P-HEG-10 suggested a perturbation in the ordering of local chain association, which could also be attributed to the hydration/swelling that may locally inhibit the chain packing at atomic length scale.”

REVIEWER COMMENTS

Reviewer #1 (Remarks to the Author):

The authors have carefully addressed all comments from the reviewers. The paper is suitable for publication in Nat Commun after minor revisions.

1. Please IRF of TCSPC data, also the fitting curves.
2. Provide relevant discussion why TEA is used, not other sacrificial donors, such as ascorbic acid or TEOA.

Reviewer #2 (Remarks to the Author):

I am generally happy with the changes made by the authors; however, some issues remain.

1. The clarification that the observed change in contact angle in Fig. 5E happens in the dark is great but I remain confused why this change happens. I am sure this might be my naivety and the fact that I have never performed the actual experiment myself but is such a change expected and why? Does this always happen or are the materials studied here special in this respect?
2. What do the authors mean when they say on page 16 that “the molecular packing of polymer in film state is severe than that in solution state”? I am not sure what severe means in this context. I could guess that the authors could mean dense but then again that makes no sense with respect to the second bit of the sentence as in the solution state the molecules are not packed (although I guess they could be coiled up).
3. I think it would be great if the authors could add the discussion of the effect of storing the material in the dark on the measured EPR spectra to the supporting information, as well as the corresponding figure used by the authors on page 17 of their reply to referees. I think this will be useful for the reader when understanding the authors’ results, as well be useful as a future reference for the community.

Reviewer #3 (Remarks to the Author):

The authors have addressed properly the issues that I raised, and the quality of the manuscript was significantly improved. It could be accepted as it is.

Reviewer #4 (Remarks to the Author):

Thank you for the response. The additional scattering measurements of the polymer and also the new study on modified PF8BT have strengthened the manuscript significantly.

One point remaining clarification is that the authors state in the response that the polymers do not absorb light at longer wavelengths. The tail at >450 nm in the absorption spectrum of the solid is not present when the polymers are in solution. But there is a non-zero quantum yield at wavelengths 450-600 nm. How can the photocatalyst be producing hydrogen if it is not absorbing at these wavelengths?

Reviewer #5 (Remarks to the Author):

The authors have addressed the reviewers' comments so I think it is fine for mostly fine for publication. I still don't fully agree with their selling point which is to stress interrupting the conjugation length. That's not the interesting part. The interesting part is that the conjugation was interrupted using hydrophilic segments so I would recommend that the authors rephrase their last sentence in the abstract. One interesting control would have been to have an equivalent polymer with the backbone interrupted with hydrophobic alkyl units but I think this is fine as is.

1. Abstract: The following statement is still unclear "Unlike the previous strategy of maintaining a long conjugation length, our approach results in DCP photocatalysts with an interrupted conjugation length that remains efficient photocatalysis." As the authors note, the effective conjugation length is not altered. As such, I would encourage the authors to be careful in the terminology used in this sentence.

2. The redrawn polymer structure is confusing. Based on IUPAC notation, the recommended way to show that you have a statistical copolymer would be to remove that large bracket labelled with subscript n, and connect the two smaller brackets with "stat" to indicate it's statistical. x and y should refer to the degree of polymerization and so if the authors want it to mean the percentage of monomers, that notation needs to be defined elsewhere.

3. I think the authors misunderstood my statement about using a solvent other than water for DLS. I think they could have found a solvent that wouldn't solubilize the polymer. Regardless, it's fine. This does not make or break the paper.

Responses to Reviewer #1

Comments to the Author

Comments:

The authors have carefully addressed all comments from the reviewers. The paper is suitable for publication in Nat Commun after minor revisions.

We thank the reviewer for your positive comments. The manuscript was carefully revised based on the suggestions and comments, and we hope that the revised manuscript is suitable for publication in this esteemed journal.

Comment 1. Please IRF of TCSPC data, also the fitting curves.

Response: We thank the reviewer for the suggestions. Per the reviewer's suggestion, we added IRF and fitting curves in Fig. 3b and Supplementary Fig. 29 of the revised manuscript.

Fig. 3 Optical and electrochemical properties of polymers. ...b Time-resolved photoluminescence spectra were obtained in suspensions (5 mg photocatalyst in 10 mL mixture consisting of equal volumes of H₂O, MeOH, and TEA). Samples were excited with a 405 nm laser and emission was measured at 455 nm. **c...**

Supplementary Fig. 29 Time-resolved photoluminescence spectra of **a** PFBPO, **b** P-EG-5, **c** P-TEG-5, **d** P-HEG-5, **e** P-HEG-10, **f** P-HEG-20, **g** P-EA-5, and **h** P-EA-10 were obtained in suspensions (5 mg photocatalyst in 10 mL solution mixture consisting of equal volumes of H₂O, MeOH, and TEA). Samples were excited with a 405 nm laser and emission was measured at 455 nm.

Comment 2. Provide relevant discussion why TEA is used, not other sacrificial donors, such as ascorbic acid or TEOA.

Response: Thank you for your valuable comments. In our previous work [*Sustainable Energy Fuels* **4**, 5264 (2020)], we have studied the optimization of different sacrificial donors. As results, PFBPO exhibited the highest HER results when using TEA as the sacrificial donor. Therefore, we used TEA as a sacrificial donor for both the PFBPO and the series of the polymers with hydrophilic non-conjugated segments in this study.

Optimization of photocatalytic activities for PFBPO using different sacrificial donors.

Responses to Reviewer #2

Comments to the Author

Comments:

I am generally happy with the changes made by the authors; however, some issues remain.

We thank the reviewer for your positive comments. The manuscript was carefully revised based on the suggestions and comments, and we hope that the revised manuscript is suitable for publication in this esteemed journal.

Comment 1. The clarification that the observed change in contact angle in Fig. 5E happens in the dark is great but I remain confused why this change happens. I am sure this might be my naivety and the fact that I have never performed the actual experiment myself but is such a change expected and why? Does this always happen or are the materials studied here special in this respect?

Response: Thanks for your valuable comments. Per the reviewer's suggestion, we further measured bare Si-wafer as a control experiment. As shown in Fig. 5, the contact angle of the bare Si-wafer dropped by less than 10 degrees over the period of 25 minutes, which confirmed that the significant changes in contact angle of PFBPO and P-HEG-10 comes from the presence of the resulting polymer. Furthermore, in order to confirm such phenomena on other materials, we further measured and compared the contact angles of PCPDTBSO and PCPDTBSO-HEG-10 polymer films. As results, in the case of the PCPDTBSO-HEG-10 film with the initial contact angle of 106° and then decreased to 69.9° over the period of 15 minutes. While for the PCPDTBSO film, the initial water contact angle of 115° decreased to 81.7° after 15 minutes. The trend change of the results are in agreement with that of PFBPO and P-HEG-10 polymer films.

Fig. 5 Photocatalytic hydrogen evolution experiments in film. a... d Correlation between the time and the water contact angles of Si-wafer, PFBPO and P-HEG-10. e...

Correlation between the time and the water contact angles of PCPDTBSO and PCPDTBSO-HEG-10.

Comment 2. What do the authors mean when they say on page 16 that “the molecular packing of polymer in film state is severe than that in solution state”? I am not sure what severe means in this context. I could guess that the authors could mean dense but then again that makes no sense with respect to the second bit of the sentence as in the solution state the molecules are not packed (although I guess they could be coiled up).

Response: Thank you for drawing our attention to this statement. To avoid the misleading description of our statement, per the reviewer’s suggestion, we modified the paragraph on page 16 of the revised manuscript.

“...Since the polymer chains in the film state are molecularly packed at a high density, resulting in that the interior of the polymer film may not be able to efficiently react with water, this result demonstrated the advantage of our main-chain engineered DCP as a thin-film photocatalyst.”

Comment 3. I think it would be great if the authors could add the discussion of the effect of storing the material in the dark on the measured EPR spectra to the supporting information, as well as the corresponding figure used by the authors on page 17 of their reply to referees. I think this will be useful for the reader when understanding the authors' results, as well be useful as a future reference for the community.

Response: Thanks for your valuable comments. Per the reviewer's suggestion, we added the discussion of the effect of storing the material in the dark on the measured EPR spectra.

As results, the corresponding figure was added in Supplementary Fig. 30 and a sentence was added in Page 8 of the revised manuscript.

“In the EPR spectra (Fig. 3c, Supplementary Fig. 30), a sharp signal was observed at 3500–3520 G and its intensity increased with additional light irradiation due to more radical generation via additional photoexcitation. The irradiation-induced signal enhancement for the DCPs was greater than that for PFBPO. Interestingly, after P-HEG-10 was stored in the dark for 3 days, although the EPR signal could still be observed, the intensity of the EPR signal was significantly decreased. Therefore, polymers already showing EPR signals without additional light irradiation can be attributed to their responded to light when stored under ambient condition. In addition, we further measured the EPR of the monomers (Supplementary Fig. 31), and found that the EPR signal of the polymer was mainly contributed by the BPO segment instead of the fluorene and HEG segments.”

Supplementary Fig. 30 EPR spectra of **a** PFBPO, **b** P-EG-5, **c** P-TEG-5, **d** P-HEG-5, **e** P-HEG-10, **f** P-HEG-20, **g** P-EA-5, and **h** P-EA-10 were measured without (dash line) and with (solid line) additional light irradiation and **i** EPR spectra of P-HEG-10 stored under ambient conditions and stored in the dark for 3 days.

Responses to Reviewer #3

Comments to the Author

Comments:

The authors have addressed properly the issues that I raised, and the quality of the manuscript was significantly improved. It could be accepted as it is.

We thank the reviewer for this positive comment and support the publication of our manuscript.

Responses to Reviewer #4

Comments to the Author

Comments:

Thank you for the response. The additional scattering measurements of the polymer and also the new study on modified PF8BT have strengthened the manuscript significantly.

Comment 1. One point remaining clarification is that the authors state in the response that the polymers do not absorb light at longer wavelengths. The tail at >450 nm in the absorption spectrum of the solid is not present when the polymers are in solution. But there is a non-zero quantum yield at wavelengths 450-600 nm. How can the photocatalyst be producing hydrogen if it is not absorbing at these wavelengths?

Response: Thank you for drawing our attention to this statement. When measuring the absorption spectra in dichloromethane solution, the polymers were completely dissolved in it. While actually, in the photocatalytic hydrogen evolution experiment, the photocatalytic solution is water/TEA/methanol, and the polymers were dispersed but not dissolved in the photocatalytic solution. Therefore, it is more reasonable to compare the AQY results with the DRS spectrum (solid state). Many previous studies also used the absorption spectra in solid-state to compare with the AQY results [*Nat. Chem.* **10**, 1180 (2018); *ACS Energy Lett.* **3**, 2544 (2018); *Angew. Chem. Int. Ed.* **58**, 10236 (2019); *Energy Environ. Sci.* **13**, 1843 (2020); *J. Am. Chem. Soc.* **142**, 11131 (2020); *Angew. Chem. Int. Ed.* **59**, 16902 (2020); *Adv. Sci.* **7**, 1902988 (2020); *Nat. Commun.* **12**, 483 (2021); *Adv. Mater.* **33**, 2008498 (2021); *Nat. Commun.* **13**, 2357 (2022); *Adv. Funct. Mater.* **32**, 2109423 (2022)]. As shown in Fig. 4d, we also compared the DRS spectra with the AQY results in this work, which is the same way as the literatures.

Responses to Reviewer #5

Comments to the Author

Comments:

The authors have addressed the reviewers' comments so I think it is fine for mostly fine for publication. I still don't fully agree with their selling point which is to stress interrupting the conjugation length. That's not the interesting part. The interesting part is that the conjugation was interrupted using hydrophilic segments so I would recommend that the authors rephrase their last sentence in the abstract. One interesting control would have been to have an equivalent polymer with the backbone interrupted with hydrophobic alkyl units but I think this is fine as is.

Thanks for the reviewer's comments. The manuscript was carefully revised based on the suggestions and comments, and we hope that the revised manuscript is suitable for publication in this esteemed journal.

Comment 1. Abstract: The following statement is still unclear "Unlike the previous strategy of maintaining a long conjugation length, our approach results in DCP photocatalysts with an interrupted conjugation length that remains efficient photocatalysis." As the authors note, the effective conjugation length is not altered. As such, I would encourage the authors to be careful in the terminology used in this sentence.

Response: Thank you for drawing our attention to this statement. To avoid a misleading description of our statement, per the reviewer's suggestion, we modified the sentence of abstract on page 2 of the revised manuscript.

"...By introducing non-conjugated hydrophilic segments, the effective conjugation length of DCP photocatalysts is not altered, resulting in that the DCPs remain efficient photocatalysis."

Comment 2. The redrawn polymer structure is confusing. Based on IUPAC notation, the recommended way to show that you have a statistical copolymer would be to remove that large bracket labelled with subscript n, and connect the two smaller brackets with “stat” to indicate it’s statistical. x and y should refer to the degree of polymerization and so if the authors want it to mean the percentage of monomers, that notation needs to be defined elsewhere.

Response: We are grateful for the reviewer’s suggestion. Per the reviewer’s suggestion, we have redrawn the chemical structures of the polymers in **Fig. 1** and **Supplementary Scheme 1** of the revised manuscript.

Fig. 1 Schematic illustration of design strategy and polymer structures. a Schematic illustration of the polymer photocatalysts containing hydrophilic non-conjugated segments. **b** The molecular structures of hydrophilic segments and the corresponding DCPs.

Supplementary scheme 1 Synthesis procedure of polymers using Pd-catalyzed Suzuki–Miyaura

coupling polymerization of fluorene-boronic ester and 3,7-dibromo-5-phenylbenzo[b]phosphindole-5-oxide with either EG-based (EG-Br, TEG-Br, and HEG-Br) or EA-based (EA-Br) segments.

Comment 3. I think the authors misunderstood my statement about using a solvent other than water for DLS. I think they could have found a solvent that wouldn't solubilize the polymer. Regardless, it's fine. This does not make or break the paper.

Response: Thank you for your valuable and insightful comments that take our manuscript to the next level.

REVIEWERS' COMMENTS

Reviewer #1 (Remarks to the Author):

The authors have well addressed the comments from me. The paper can be accepted.

Reviewer #2 (Remarks to the Author):

I am happy with the changes made and have no further comments.

Reviewer #4 (Remarks to the Author):

All fine, one minor point ("severe packing") is still a phrase used in the text that could be clarified.

Reviewer #5 (Remarks to the Author):

I am satisfied with the changes made by the authors.

National Tsing Hua University

Department of Chemical Engineering

No. 101, Sec. 2, Kuang-Fu Road
Hsinchu 300044, Taiwan

Responses to Reviewer #1

Comments to the Author

Comments:

The authors have well addressed the comments from me. The paper can be accepted.

We thank the reviewer for this positive comment and support the publication of our manuscript.

Responses to Reviewer #2

Comments to the Author

Comments:

I am happy with the changes made and have no further comments.

We thank the reviewer for this positive comment and support the publication of our manuscript.

Responses to Reviewer #4

Comments to the Author

Comments:

All fine, one minor point ("severe packing") is still a phrase used in the text that could be clarified.

We thank the reviewer for this positive comment and support the publication of our manuscript.

To avoid the misleading description of our statement, per the reviewer's suggestion, we modified the paragraph on page 10 of the revised manuscript.

"...We considered that the high density packing of polymer chains in the films results in non-obvious differences in water contact angles between DCPs and PFBPO."

Responses to Reviewer #5

Comments to the Author

Comments:

I am satisfied with the changes made by the authors.

We thank the reviewer for this positive comment and support the publication of our manuscript.